# Stage-dependent remodeling of projections to motor cortex in ALS mouse model revealed by a new variant retrograde-AAV9

Barbara Commisso[1], Lingjun Ding[1], Karl Varadi[2], Martin Gorges[1], David Bayer[1], Tobias M Boeckers[3], Albert C Ludolph[1], Jan Kassubek[1], Oliver J Müller[4], Francesco Roselli[1,3]*

[1]Department of Neurology, University of Ulm, Ulm, Germany; [2]Department of Internal Medicine III, University Hospital Heidelberg, Heidelberg, Germany; [3]Department of Anatomy and Cell biology, University of Ulm, Ulm, Germany; [4]Department of Internal Medicine, University of Kiel, Kiel, Germany

*For correspondence:
francesco.roselli@uni-ulm.de

Competing interests: The authors declare that no competing interests exist.

**Abstract** Amyotrophic lateral sclerosis (ALS) is characterized by the progressive degeneration of motoneurons in the primary motor cortex (pMO) and in spinal cord. However, the pathogenic process involves multiple subnetworks in the brain and functional MRI studies demonstrate an increase in functional connectivity in areas connected to pMO despite the ongoing neurodegeneration. The extent and the structural basis of the motor subnetwork remodeling in experimentally tractable models remain unclear. We have developed a new retrograde AAV9 to quantitatively map the projections to pMO in the SOD1(G93A) ALS mouse model. We show an increase in the number of neurons projecting from somatosensory cortex to pMO at presymptomatic stages, followed by an increase in projections from thalamus, auditory cortex and contralateral MO (inputs from 20 other structures remains unchanged) as disease advances. The stage- and structure-dependent remodeling of projection to pMO in ALS may provide insights into the hyperconnectivity observed in ALS patients.
DOI: https://doi.org/10.7554/eLife.36892.001

## Introduction

Amyotrophic lateral sclerosis (ALS) is classically described as a disease of upper and lower motor neurons (*Ravits et al., 2007*). However, postmortem studies have demonstrated that pathological hallmarks of ALS (such as TDP-43 inclusions) appear in motor cortex as well as in multiple cortical and subcortical structures during disease progression, depicting a propagation pattern (*Brettschneider et al., 2013*; *Braak et al., 2013*). Thus, ALS has been growingly conceptualized as a disease of the motor subnetwork or, in more advanced stages, a multi-network disease affecting motor, premotor and sensory areas. To which extent these networks are damaged by the neurodegeneration and remodel themselves during the disease progression, it is a subject of active investigation.

Diffusion-tensor imaging (DTI) studies have revealed that the degeneration of axons is detected in more and more white-matter tracts affecting different structures (corticospinal, corticorubral, corticostriatal and perforant tracts) at different stages of the propagation scheme (*Neuroimaging Society in ALS (NiSALS) DTI Study Group et al., 2016*; *Kassubek et al., 2014*; *Kassubek et al., 2018*) as well as in interhemispheric connections (*Filippini et al., 2010*) and in spinal cord (*Cohen-Adad et al., 2013*). However, analysis of resting-state networks by functional

magnetic resonance imaging (fMRI) has revealed increased functional connectivity has been described in symptomatic ALS patients in the sensorimotor (*Menke et al., 2016*; *Agosta et al., 2018*) somatosensory (*Agosta et al., 2011*) and cortico-striatal networks (*Fekete et al., 2013*). Notably, an increase in functional connectivity in the motor subnetwork was already detectable in presymptomatic and early-stage ALS patients (*Schulthess et al., 2016*).

Although these data indicate that a significant architectural remodeling of the motor subnetwork takes place in ALS patients in the face of ongoing neurodegeneration, the extent of such remodeling and the structural changes underlying this effect in the motor network are unclear. Of particular interest is the extent of remodeling of the input network to pMO: in fact, the type and amount of projection to pMO determines not only the output from MO, which is strongly related to the execution of movements; in addition, the synaptic inputs are strong regulators of neuronal activity pattern and may influence activity-dependent pathogenic factors (*Roselli and Caroni, 2015*; *Bading, 2017*). To date, connectivity maps have been obtained at mesoscale (*Oh et al., 2014*; *Mitra, 2014*) or at cellular resolution (*Mao et al., 2011*; *Wertz et al., 2015*) only for wild-type models.

In order to address the remodeling of the subnetwork in an experimentally tractable condition, we have selected to quantitatively map the projections to the primary motor cortex network in the SOD1(G93A) ALS mouse model using a newly developed AAV variant endowed with retrograde tracing abilities. We have demonstrated that selected cortical areas display an increased projection to pMO already at presymptomatic stage and the increase in projection expands during disease progression to involve additional cortical and subcortical structures. We show that the pattern observed in the mouse model bears significant similarities with fMRI functional connectivity data gathered from human ALS patients. Thus, we have identified one structural component of the early and selective remodeling of the large-scale architecture of the motor subnetwork, which may contribute to explain the functional connectivity changes observed in patients.

## Results

### AAV9-SLR variant is a new, efficient viral tool for retrograde connectivity tracing

In order to identify a variant of AAV9 suitable for retrograde tracing, we considered three variants previously generated (AAV9-SLR, AAV9-NSS and AAV9-RGD) as part of an ongoing screening of the properties of newly generated AAV9 variants. These variants displayed a high infection efficiency in vitro in multiple cell lines (human coronary artery endothelial cells (HCAECs), human coronary artery smooth muscle cells (HCASMC), HEK293T, HeLa, 911, HepG2; *Varadi et al., 2012*), suggesting a potential for infecting neurons and neuronal processes with high efficiency; in particular, we evaluated them for highly desired applications such as high local infectivity and retrograde infectivity. We first tested infection rate of these AAV9 variants in vivo (*Figure 1A*). Five independent groups of tdTomato-ROSA26 reporter mice (N = 3 for each group, age P20) were injected in the dorsal striatum (DS) with 500 nl of viral suspension ($9*10^{13}$ genomes/ml) of AAV9-SLR, AAV9-NSS, AAV9-RGD and, for comparison, WT-AAV9 and WT-AAV2 (*Figure 1B*). The total number of infected neurons in the injection area (local infectivity) and in regions projecting to DS was assessed at 15 days post-injection (DPI). One-way ANOVA revealed a significant difference ($F_{(4,10)}=198.8$, p<0.0001) in the number of infected neurons per area unit (infection rate) in the injection ROI. Compared to WT-AAV9, infection rate was significantly higher for AAV9-SLR (post-hoc p=0.0001) but not for AAV-NSS, (p=0.0910), and was significantly lower for both AAV-RGD (p=0.0001) and WT-AAV2 (p=0.0001; detailed statistics are reported in *Supplementary file 1a*). In agreement with the limited toxicity of AAVs, no morphological sign of distress (axonal beading, dendrites fragmentation, fluorescent fragments of cell bodies) were identified in neurons infected with the AAV variants or with the WT serotypes.

When screened for their ability to retrogradely infect neurons projecting to the injection site (in DS), the AAV variants differed significantly from WT-AAV9 in terms of retrogradely infected neurons located in substantia nigra (SNc; one-way ANOVA $F_{(4,10)}=135.1$, p>0.0001; detailed statistics are reported in *Supplementary File 1b*) and motor cortex (MO; $F_{(4,10)}=46.5$, p<0.0001; *Supplementary File 1c*). Injection of AAV9-SLR and AAV9-NSS resulted in a larger number of tdTomato+/tyrosine-hydroxylase-positive (TH+) neurons in SNc than WT-AAV9 (AAV9-SLR: p=0.0001;

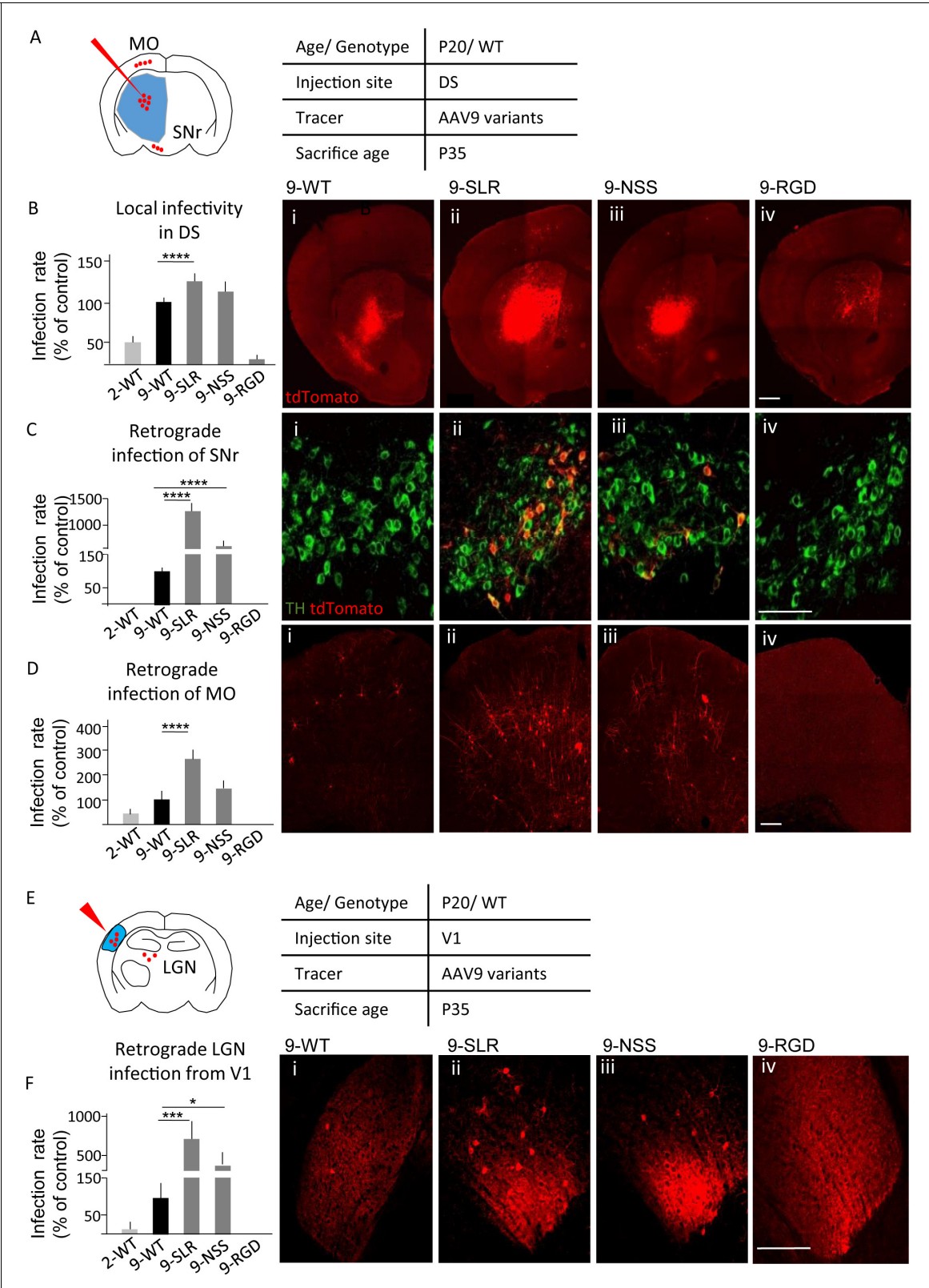

**Figure 1.** AAV9-SLR is a new AAV variant with pronounced retrograde infectivity. AAV9-SLR, AAV9-NSS and AAV9-RGD variants (or WT-AAV9 and WT-AAV2) were injected in dorsal striatum (N = 3) and local infection rate (B), retrograde infection to Substantia Nigra (C) and Motor Cortex (D) were assessed. (A) Experimental design showing injection site in dorsal striatum (DS) and brain regions that have been investigated for retrograde infection (SNr and MO). (B) AAV9-SLR displayed a significantly higher local infection rate than WT-AAV9 (143 ± 10%; post-hoc p=0.0001, whereas rAAV-NSS was

*Figure 1 continued on next page*

*Figure 1 continued*

comparable to WT-AAV9 (114 ± 8%, p=0.0910) and AAV-RGD was significantly less effective (8 ± 1%, p=0.0001). Detailed statistic provided in *Supplementary file 1a*. (B i-iv) representative images of injection site in DS. Scale bar 500 μm. (C): after DS injection, rAAV9-SLR produced a ten-fold increase in retrogradely infected TH +neurons compared to WT-AAV9 (1267 ± 152%, p=0.0001); rAAV9-NSS was more effective than WT-AAV9 (600 ± 100%, p=0.0001) but less effective than AAV9-SLR. WT-AAV2 and AAV-RGD did not retrogradely infect TH+ neurons. Detailed statistic provided in *Supplementary file 1b*. (C i-iv) representative images of SNr. Scale bar 30 μm. (D) after DS injection, rAAV9-SLR displayed the highest rate of retrograde infection in motor cortex (236 ± 32% of WT-AAV9, p=0.0001), followed by rAAV9-NSS (145 ± 25% p=0.0909), both more effective than WT-AAV9. Detailed statistic provided in *Supplementary file 1c*. (D i-iv) representative images of motor cortex. Scale bar 100 μm. (E) Experimental design showing injection site in primary visual cortex (V1) and brain regions that have been investigated for retrograde infection (LGN). After injection in primary visual cortex injection (V1, N = 3), retrograde infection of neurons in lateral geniculate nucleus (LGN) was assessed. (F) rAAV9-SLR displayed a strong retrograde infection of LGN neurons (700 ± 240%, p=0.0005), followed by AAV9-NSS (400 ± 133% of WT-AAV9, p=0.0446). No LGN infection resulted from the injection of WT-AAV2 or AAV9-RGD. Detailed statistic provided in *Supplementary file 1d*. (F i-iv) Representative images of LGN. Scale bar 30 μm. Manual cell counting was normalized on the volume of the injection sites, values are expressed in terms of percentage over AAV9-WT. Ordinary one-way ANOVA with Dunnett multiple comparison test was used.

DOI: https://doi.org/10.7554/eLife.36892.002

The following source data is available for figure 1:

**Source data 1.** Detailed statistic concerning AAV variants infectivity (*Figure 1*).

DOI: https://doi.org/10.7554/eLife.36892.003

AAV9-NSS: p=0.0001; *Figure 1C*), whereas injection of WT-AAV2 and AAV-RGD resulted in no retrograde infection to SNc. Increased retrograde infection from DS to MO (*Figure 1D*) was detected for AAV9-SLR (p=0.0001) but not for AAV9-NSS (p=0.0909), whereas WT-AAV2 (p=0.1153) and AAV9-RGD (p=0.0018) had a significantly lower retrograde infection rate. Finally, we evaluated the retrograde infection rate from visual cortex (V1) to lateral geniculate nucleus (LGN, *Figure 1E*). Injection of 500 nl of suspension of each of the five AAV variants in V1 of reporter tdTomato/ROSA26 (N = 3) resulted in a significantly different number of tdTomato + neurons in LGN (one-way ANOVA, $F_{(4,10)}= 17.8$ p=0.0002; *Figure 1F*). AAV9-SLR and AAV9-NSS infected a significantly larger number (up to 10-fold) of LGN neurons than WT-AAV9 (AAV9-SLR p=0.0005; AAV9-NSS: p=0.0446; detailed statistics are reported in *Supplementary file 1d*), whereas, neither WT-AAV2 (p=0.7941) nor AAV9-RGD (p=0.7271) resulted in any infection in the LGN. Taken together, these data identified AAV9-SLR as a new AAV variant endowed with robust, reproducible and broad retrograde infection capability.

## Mapping input to primary motor cortex by AAV9-SLR

We then exploited the AAV9-SLR for mapping the forebrain and subcortical neurons projecting to primary (pMO, *Figure 2A*) or secondary (sMO) motor cortex in WT mice. Retrogradely labeled neurons in the cortical and subcortical structures of the forebrain were quantified, anatomically annotated and normalized for the volume of the injection site (*Figure 2B,C*). Overall, we identified 28 distinct anatomical structures providing direct input to the injection site in pMO, including thalamic nuclei (TH; *Figure 2D*), followed by homolateral somatosensory cortex (SS; *Figure 2D*), contralateral secondary motor cortex (cMOs) and contralateral primary motor cortex (cMOp; *Figure 2D*). Values for raw neuronal counts (before normalization) are displayed in *Figure 2—figure supplement 1A*, detailed statistic provided in *Supplementary file 1e*. Detailed statistic after normalization for injection site volume are in *Supplementary file 1f*. When projecting neurons are expressed in terms of percentage of the total population of projecting neurons (*Figure 2—figure supplement 1B*), TH represented 32 ± 7% of the total input to pMO, SS 22 ± 7%, cMOs 8 ± 3% and cMOp 6 ± 2% (full details are provided for all structures in *Supplementary file 1g*). Additional projections were provided by auditory cortex (AUD 6 ± 1%; *Figure 2D*) and contralateral somatosensory cortex (cSS 5 ± 5%) and, among subcortical structures, by caudoputamen (CP 6 ± 2%), claustrum (CLA 3 ± 1%) and basolateral amygdala (BLA 0.4 ± 0.3%). We consistently identified a small contingent of neurons projecting to pMO located in hypothalamus (HY 0.6 ± 0.2% of the total pool of neurons projecting to pMO in the forebrain); out of these hypothalamic neurons, not previously reported, the majority was located in the lateral zone of hypothalamus (zona incerta 23 ± 10% of the total of hypothalamic neurons projecting to pMO; lateral preoptic area 23 ± 19% of the total of hypothalamic neurons projecting to pMO; lateral hypothalamic area 14 ± 18% of the total of hypothalamic neurons projecting to

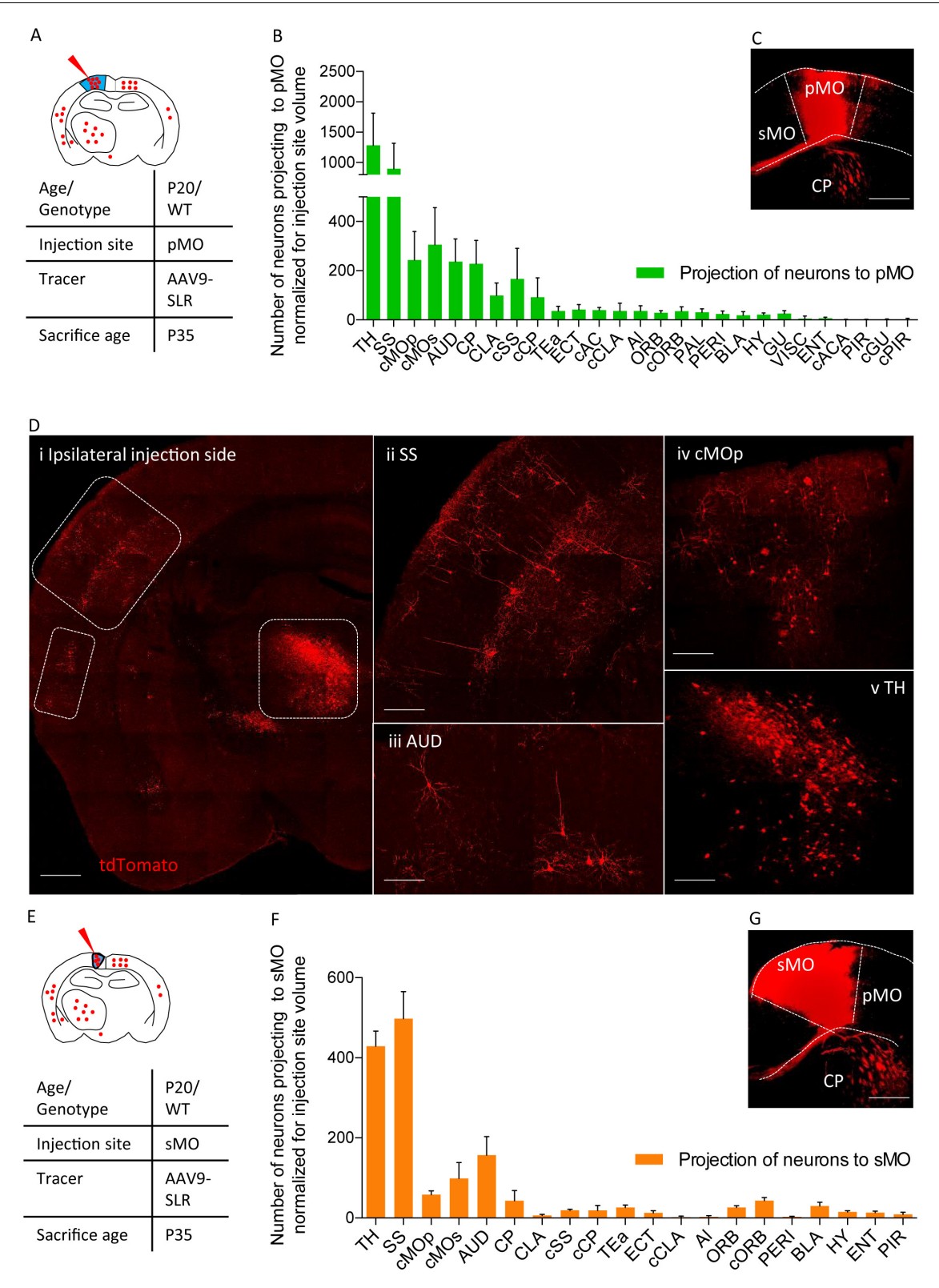

**Figure 2.** Projections to primary and secondary motor cortex in P20 wild-type. (**A**) Experimental design depicting injection site in pMO and relative projecting neurons. (**B**) List of the regions projecting to primary MO (pMO) identified by injection of AAV9-SLR in pMO of WT mice (representative injection site in panel (**C**) scale bar 300 µm). Total neuronal count has been normalized for the volume of the injection site (raw neuronal counts in *Supplementary file 1e*; detailed statistic on normalized counts provided in *Supplementary file 1f*;). The largest contribution to the input to pMO was

*Figure 2 continued on next page*

*Figure 2 continued*

provided by thalamus (TH), followed by somatosensory (SS), contralateral secondary motor cortex (cMOs) and contralateral primary motor cortex (cMOp). Additional inputs came from: auditory cortex (AUD), caudoputamen (CP), contralateral SS (cSS), CLA, baso-lateral amygdala (BLA) and hypothalamus (HY). List of the regions projecting to primary motor cortex in terms of total neuronal count and in term of percentage over the total pool of positive neurons in *Figure 2—figure supplement 1A–B*, detailed statistic provided in *Supplementary file 1e and 1g*. (D) Representative images of retrogradely-labelled neurons projecting to pMO. (i) Ipsilateral brain section displaying main regions targeting primary motor cortex. Scale bar 500 µm. (ii) Detail of SS, scale bar 200 µm. (iii) Detail of AUD, scale bar 200 µm. (iv) Detail of cMOp, scale bar 200 µm. (v) Detail of TH, scale bar 200 µm. (E) Experimental design depicting injection site in sMO and relative projecting neurons. (F): List of the regions projecting to secondary MO (sMO) identified by injection of AAV9-SLR in sMO of WT mice (representative injection site in panel (G) scale bar 300 µm). Total neuronal count has been normalized for the volume of the injection site (detailed statistic provided in *Supplementary file 1e-1g*). Input toward pMO and sMO is qualitatively similar (display of total neuronal count in *Figure 2—figure supplement 1C*, detailed statistic provided in *Supplementary file 1e-1g*). List of the regions projecting to secondary motor cortex in terms of percentage over the total pool of positive neurons in *Figure 2—figure supplement 1D*, detailed statistic provided in *Supplementary file 1f*. Projections from thalamus, to both pMO and sMO, have been further broken down for each thalamic nucleus (*Figure 2—figure supplement 2A–D*, detailed statistic provided in *Supplementary file 1h-1i-1j*). Abbreviations: Thalamus (TH), homolateral somatosensory cortex (SS), contralateral secondary motor cortex (cMOs), contralateral primary motor cortex (cMOp), auditory cortex (AUD), caudoputamen (CP), contralateral somatosensory cortex (cSS), claustrum (CLA), contralateral caudoputamen (cCP), temporary association area (TEa), ectorinal cortex (ECT), contralateral anterior cingulate (cAC), contralateral claustrum (cCLA), agranular insular area (AI), orbital cortex (ORB), contralateral orbital cortex (cORB), paraventricular hypothalamic area (PAL), perirhinal area (PERI), basolateral amygdala (BLA), hypothalamus (HY), gustatory area (GU), visceral area (VISC), entorhinal area (ENT), contralateral anterior part of anterior commissure (cACA), endopiriform nucleus (EP), piriform area (PIR), contralateral gustatory area (cGU), contralateral piriform area (cPIR).

DOI: https://doi.org/10.7554/eLife.36892.004

The following source data and figure supplements are available for figure 2:

**Source data 1.** Detailed statistic concerning projecting neurons to primary motor cortex (pMO) and secondary motor cortex (sMO) in WT animals(P20) traced via AAV9-SLR injection.

DOI: https://doi.org/10.7554/eLife.36892.009

**Figure supplement 1.** Projections to primary and secondary motor cortex in P20 wild-type.

DOI: https://doi.org/10.7554/eLife.36892.005

**Figure supplement 1—source data 1.** Detailed statistic concerning projecting neurons to primary motor cortex (pMO) and secondary motor cortex (sMO) in WT animals (P20) traced via AAV9-SLR injection.

DOI: https://doi.org/10.7554/eLife.36892.006

**Figure supplement 2.** Projection from thalamic nuclei to primary and secondary motor cortex in WT at P20.

DOI: https://doi.org/10.7554/eLife.36892.007

**Figure supplement 2—source data 1.** Detailed statistic concerning input from thalamic nuclei to primary motor cortex (pMO) and secondary motor cortex (sMO) in WT animals(P20) traced via AAV9-SLR injection.

DOI: https://doi.org/10.7554/eLife.36892.008

---

pMO). In addition to previously reported input structures to pMO (*Hooks et al., 2013*; *Oh et al., 2014*; *Mao et al., 2011*), we identified additional inputs from paraventricular hypothalamic area (PAL $0.8 \pm 0.2$) and gustatory area (GU $0.7 \pm 0.3\%$).

The input to the neighboring sMO was qualitatively similar to pMO, including all of the structures projecting to pMO (all in all, 20 distinct regions were found to project to sMO compared to 28 for pMO). Values for raw neuronal counts (before normalization) are displayed in *Figure 2—figure supplement 1C*, detailed statistic provided in *Supplementary file 1e*). The trend remained unchanged after normalization for injection site volume (*Figure 2E–G*; detailed statistic *Supplementary file 1f*). When projecting neurons are expressed in terms of percentage of the total population of projecting neurons (*Supplementary file 1*, *Figure 2D*), SS emerged as contributing a significantly larger share of input to sMO than to pMO ($33 \pm 3\%$, p<0.0001 vs pMO), followed by structures whose input to pMO and sMO was comparable (detailed values reported in *Supplementary file 1g*): TH ($28 \pm 1\%$, p=0.5965), AUD ($10 \pm 2\%$, p=0.1914), cMOs ($7 \pm 3\%$, p>0.9999), cMOp ($4 \pm 1\%$, p=0.9815), orbital area (ORB: $2 \pm 0.4\%$, p>0.9999) and BLA ($2 \pm 0.7\%$, p>0.9999).

Since the thalamus appeared to provide a significant fraction of the projections to pMO and sMO, we annotated the labelled neurons to thalamic nuclei (raw and normalized neuronal counts are reported in *Supplementary file 1h and 1i*, respectively). In total, 19 thalamic nuclei were found to project to either pMO or sMO motor cortex (*Figure 2—figure supplement 2A–F*). When considered as a fraction of the total number of thalamic neurons projecting to pMO or sMO, the ventral anterior-lateral complex and the ventral medial nucleus appeared to be the most relevant source of

projection to MO, and their contribution to pMO or sMO was comparable (VAL to pMO 29 ± 6%, to sMO 34 ± 6%, two-way ANOVA p=0.6496) and ventral medial nucleus (VM to pMO 24 ± 7%, to sMO 25 ± 12%, p>0.9999; detailed values reported in *Supplementary file 1j*).

## Increased input to pMO in early presymptomatic mSOD

We then exploited the retrograde AAV9-SLR to quantitatively explore the projections to pMO in mSOD mice and WT littermates during disease progression. First, we focused on early-presymptomatic stages: AAV9-SLR was injected in pMO at P20 and animals were sacrificed at P35 (*Figure 3A*). Retrogradely labeled neurons in the cortical and subcortical structures of the forebrain were quantified, anatomically annotated and the raw counts normalized according to the volume of the injection site (*Figure 3B,C*). Strikingly, we found an overall increase in the number of neurons projecting to pMO in mSOD compared to WT both in raw counts (*Supplementary file 2a* and *Figure 3—figure supplement 1A*) and after normalization (Two-way ANOVA, brain regions $F_{(26,189)}$=53.0 p<0.0001, genotype $F_{(1,189)}$=16.3 p<0.0001). Post-hoc analysis (*Figure 3B*; detailed numerical values in *Supplementary file 2b*) revealed a statistically significant increase in the mean number of neurons projecting to pMO from SS (effect size: 1.6; p<0.0001) and from cMOs (effect size: 1.6; p=0.0189). Notably, among the 28 anatomical regions considered, all others displayed a comparable number of neurons projecting to pMO in WT and mSOD (p>0.05 WT vs mSOD for all the brain regions). For all regions considered, the median value of neurons projecting to pMO was close to the mean value, and the skewness value never exceeded 1.7 (with 16/28 regions in WT and 28/28 regions in mSOD having values ranging between −1.0 and +1.0), discounting the possibility of strongly biased or non-normal distributions.

We further investigated the pattern of neurons projecting to pMO from structures with increased projection (SS, cMOs) and unaltered projection (cMOp, AUD) annotating them by cortical layers (detailed values before and after normalization for injection site volume and statistical analysis reported in *Supplementary file 2c and d*). Interestingly, despite the overall increase in the number of neurons projecting to pMO, a significant loss of projecting neurons was detected in SS layer V of mSOD mice (effect size: −1.5; p=0.0324, *Supplementary file 2a*) mirrored by a reciprocal trend toward increase projection from layer II/III (effect size: 1.2; p=0.2734) but not in layer VI (p=0.9898). (*Figure 3D*; values expressed in terms of percentage of the total population of projecting neurons, detailed statistic *Supplementary file 2e*). On the other hand, the distribution of projecting neurons across cortical layers was comparable in WT and mSOD for cMOs, cMOp and AUD (data before normalization displayed in *Figure 3—figure supplement 1B*).

To rule out any infectivity bias of the virus in WT vs mSOD mice, we verified that the selective increase in projection from SS to pMO could be detected by an independent method. We injected pMO with fluorescently labeled cholera toxin B (CTb) and assessed the number of CTb-labeled neurons in SS and cMOp (showing increased or unchanged projection to pMO, respectively, in the viral tracing experiment); neuronal counts were normalized for injection volume (*Figure 4A–C*; counts before normalization are depicted in *Figure 4—figure supplement 1A*, and numerical values in *Supplementary file 2f*). Confirming the viral tracing data, CTb labeled a larger number of neurons in the SS in the mSOD compared to WT (both N = 3; one-way ANOVA $F_{(3,8)}$=17.0, p=0.0008), whereas no difference was detected in cMOp between the two genotypes (SS effect size: 1.4; p=0.0156, cMOp effect size: −0.4; p=0.9971; *Figure 4A–C* and *Supplementary file 2g*).

We further sought to rule out any differential infectivity of the virus in mSOD model by quantifying the neurons projecting to primary visual cortex (V1) from LGN and MO in P20 mice (*Figure 4D–F*). Injection of AAV9-SLR in V1 resulted in a comparable number of labelled neurons (either before either after normalization for the injection site volume: *Supplementary file 2h and 2i*) in WT (N = 3) and mSOD (N = 3) both in LGN (p=0.9263) and in MO (p=0.6360). Counts before normalization are depicted in *Figure 4—figure supplement 1B*.

In contrast to our observation in the pMO, input to sMO was comparable in WT (N = 3) and mSOD (N = 4) mice (either before either after normalization; for normalized two- way ANOVA, genotype $F_{(1,100)}$=0.7572 p=0.3863. *Figure 5A–C* and *Supplementary file 2j and 2k* for non-normalized and normalized counts, respectively). After normalization, a trend toward increased projection to sMO in mSOD was detected only in cMOs (effect size: 1.3; p=0.0708). In conclusion, retrograde tracing of the input to pMO revealed an unexpected increase in the number of neurons projecting from

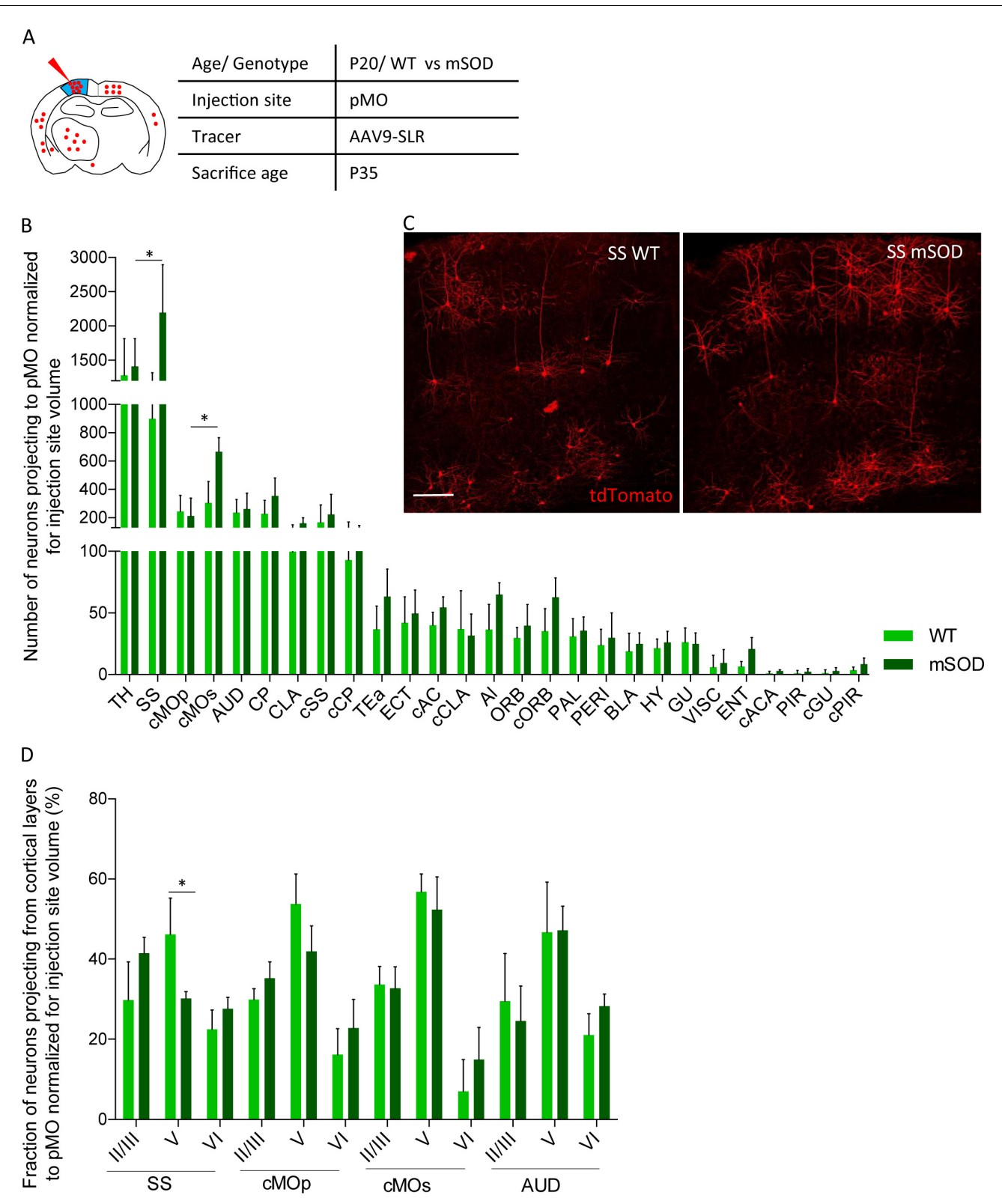

**Figure 3.** Increased projections to primary motor cortex in mSOD at pre-symptomatic stage (P20) from ipsilateral SS and contralateral sMO. (**A**) Experimental design depicting injection site in pMO and relative projecting neurons. (**B**) Number of neurons, normalized for injection site volume, projecting to primary motor cortex in WT (N = 6) and mSOD (N = 3). Significant increase in the number of neurons projecting to pMO from SS (WT vs mSOD, p<0.0001) and from cMOs (WT vs mSOD, p=0.0189). No other statistical significant differences between WT and mSOD among the 28

*Figure 3 continued on next page*

*Figure 3 continued*

identified structures. Detailed statistic provided in **Supplementary file 2b**. Input to pMO expressed in terms of total neuronal count in **Figure 3— figure supplement 1A**, detailed statistic provided in **Supplementary file 2a**. (C) Representative images of neurons labeled by retrograde tracing from SS, in WT and mSOD, respectively. Scale bar 150 μm. (D) Cortical layer allocation of neurons projecting to primary motor cortex from SS, cMOp, cMOs and AUD out of AAV9-SLR injection. Despite the overall increase in projection from SS to pMO in mSOD mice, a significant loss of projections from layer V of SS was detected (WT 46 ± 9% vs mSOD 30 ± 2%, p=0.0324). Detailed statistic provided in **Supplementary file 2c**. Values are expressed in term of percentage over the total number of neurons for each structure. Bar chart displaying cortical projections in terms of total neuron in **Figure 3— figure supplement 1B**, detailed statistic is provided in **Supplementary file 2c and 2d**. Two-way ANOVA with Sidak correction for multiple comparison.
DOI: https://doi.org/10.7554/eLife.36892.010

The following source data and figure supplements are available for figure 3:

**Source data 1.** Detailed statistic concerning projecting neurons to primary motor cortex (pMO) in WT vs mSOD animals(P20) traced via AAV9-SLR injection.
DOI: https://doi.org/10.7554/eLife.36892.013

**Figure supplement 1.** Increased projections to primary motor cortex in mSOD at pre-symptomatic stage (P20) from ipsilateral SS and contralateral sMO.
DOI: https://doi.org/10.7554/eLife.36892.011

**Figure supplement 1—source data 1.** Detailed statistic concerning projecting neurons to primary motor cortex (pMO) in WT vs mSOD animals (P20) traced via AAV9-SLR injection.
DOI: https://doi.org/10.7554/eLife.36892.012

SS and cMOs to pMO as early as P20. This reveals that remodeling of cortical connectivity is already initiated in presymptomatic mSOD mice.

## Cortical and subcortical structures display an increased input to pMO in mSOD during disease progression

We then investigated how the projections to pMO changed during disease progression. We compared the number and anatomical locations of neurons projecting to pMO in mSOD and WT mice at the early-symptomatic age, P60 (WT N = 3, mSOD N = 4) and symptomatic age, P90 (WT N = 5, mSOD N = 5; *Figure 6A, B*); investigation of later timepoints was prevented by the observation, in a preliminary study, of high intraoperative mortality (3 out of 3 mSOD mice, possibly due to respiratory failure) in mice injected at P115. The number of neurons projecting to pMO from the different structures was expressed as fold-change of the age-match WT (detailed values reported in *Supplementary file 3a*). Distribution of the projections to pMO was stable over time in WT mice (two-way ANOVA; brain regions $F_{(19,209)}=0.001$ p>0.99). However, when compared to their age-matched WT littermates at P60, mSOD mice displayed an increase in neurons projecting to pMO (two-way ANOVA; brain regions $F_{(19,100)}=2.3$ p=0.0034) from SS (effect size: 1.6; p=0.0019; unchanged compared to P20 mSOD, p>0.9999) and cMOs (effect size: 1.6; p=0.0454; unchanged compared to P20 mSOD, p>0.9999). In addition, now an increased number of neurons projecting to pMO was found in cMOp (effect size: 1.4; p=0.0077 in mSOD vs WT littermates,+2.4 ± 0.9 vs P20 mSOD, p=0.0110). Besides these three, no other structure, among the 28 analyzed, showed a significant difference between WT and mSOD (*Figure 5A–B* and *Supplementary file 3a*). Investigation of the symptomatic stage (P90) revealed a further, significant modification of the abnormal pattern of projections to pMO (two-way ANOVA; brain regions $F_{(19, 160)}=7.6$, p<0.0001), affecting both cortical and subcortical regions. Projections to pMO from SS was still increased (effect size:1.6; p=0.0095; same increase registered for P20 and P60 mSOD, p>0.9999 for both), cMOp (effect size: 1.7; p<0.0001, also increased when compared to the P60 timepoint:+1.6 ± 0.5 vs P60 mSOD, p=0.0124) and cMOs (Effect size: 1.4; p<0.0001, comparable to the increase registered in mSOD at P20 (p=0.7658) and P60 (p=0.3917)). Moreover, increased number of projecting neurons was detected from TH (effect size: 1.7; p=0.0014;+1.8 ± 0.4 vs P20 mSOD, p=0.0064), AUD (effect size: 1.6; p<0.0001;+1.4 ± 0.5 vs P20 mSOD, p<0.0001), and CP (effect size: 1.7; p<0.0001;+2.0 ± 0.5 vs P20 mSOD, p=0.0086). As for the P20 timepoint, median values were close to the mean value of relative change in the number of projecting neurons and skewness ranged between −1.0 and +1.0, indicating the absence of biased distributions of the data.

To exclude an effect of a widespread neuronal loss in the pMO of mSOD on the number of axons projecting to pMO, we performed a NeuN staining in the pMO of P90 mSOD mice (n = 3) revealing

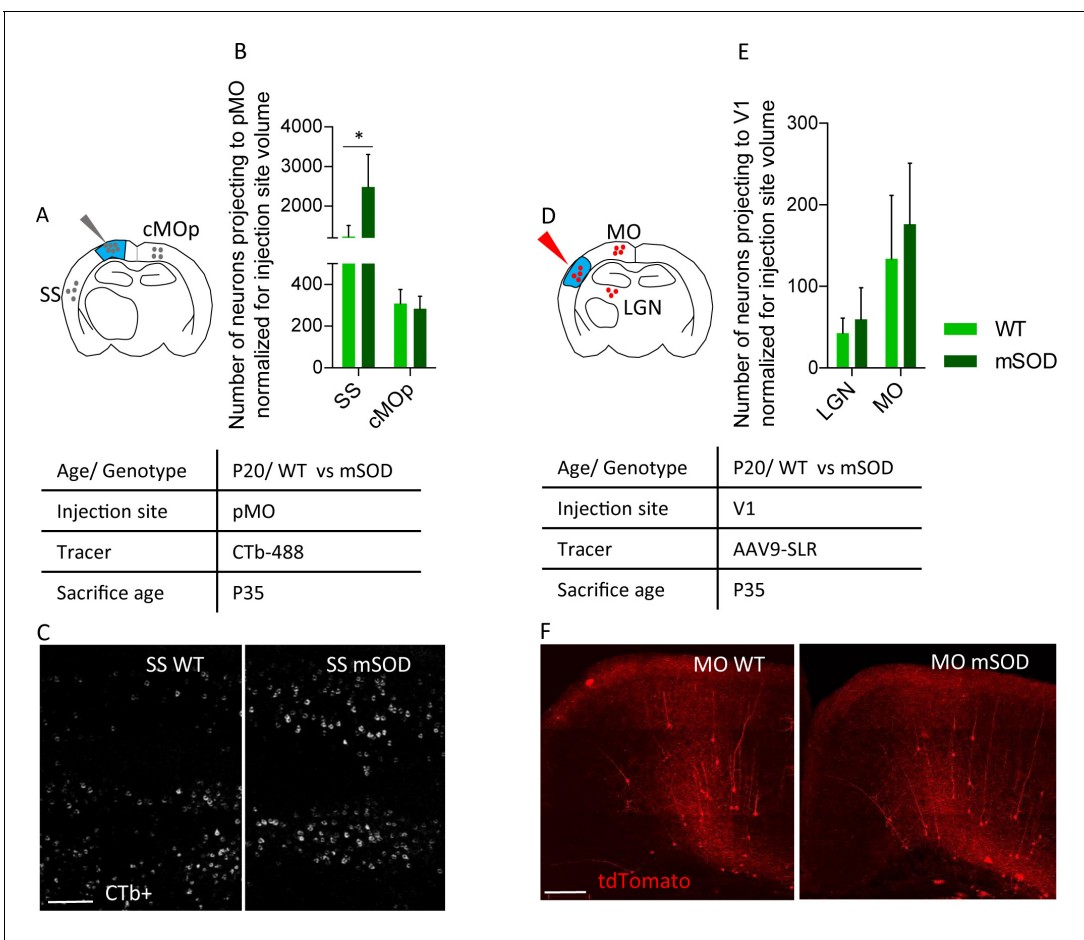

**Figure 4.** Increased projections to primary motor cortex in mSOD at pre-symptomatic stage (P20) is specific for motor network and is confirmed via the retrograde tracer choleratoxin. (**A**) Experimental design depicting injection site in pMO via choleratoxin (CTb) with two regions that have been investigated for retrograde infection (SS and cMOp). (**B**) Normalized (for injection site volume) number of neurons projecting to primary motor cortex, WT (N = 3) vs mSOD (N = 3), via Choleratoxin-b injection. CTb + retrogradely labeled projections from SS and cMOp to pMO. Increased projections from SS to pMO was again detected in mSOD (WT 1235 ± 280; mSOD 2487 ± 819, p=0.0156); projections from cMOp were comparable (WT 308 ± 67, mSOD 283 ± 59, p=0.9971), as in the virus tracing experiment. Detailed statistic provided in *Supplementary file 2g*. Display of total neuronal count in *Figure 4—figure supplement 1A*, detailed statistic for total neuronal count in *Supplementary file 2f*. (**C**) Representative images of SS from WT and mSOD, respectively, displaying CTb + neurons, scale bar 200 µm. (**D**) Experimental design depicting injection site in in primary visual cortex (V1) via AAV9-SLR with two regions that have been investigated for retrograde infection (LGN and MO). (**E**) Normalized (for injection site volume) number of neurons projecting to visual network (V1) from LGN (WT 42 ± 18 vs mSOD 60 ± 38; p=0.9263) and MO (WT 134 ± 77 vs SOD 176 ± 74, p=0.6360) in WT (N = 3) and mSOD (N = 3). No difference in projections from the two areas to V1 was found between WT vs mSOD. Detailed statistic provided in *Supplementary file 2i*. Display of total neuronal count in *Figure 4—figure supplement 1B*, detailed statistic in *Supplementary file 2h*. (**F**) Representative images of MO projection neurons to V1 in WT vs mSOD; scale bar 150 µm. Ordinary one-way ANOVA with Dunnett multiple comparison test.

DOI: https://doi.org/10.7554/eLife.36892.014

The following source data and figure supplement are available for figure 4:

**Source data 1.** Total number of neurons projecting to primary motor cortex, WT vs mSOD via Choleratoxin-b injection.

DOI: https://doi.org/10.7554/eLife.36892.016

**Figure supplement 1.** Increased projections to primary motor cortex in mSOD at pre-symptomatic stage (P20) is specific for motor network and is confirmed via the retrograde tracer choleratoxin.

DOI: https://doi.org/10.7554/eLife.36892.015

no difference in the total number of neurons in our injection area (p=0.4987, *Figure 6—figure supplement 1A*).

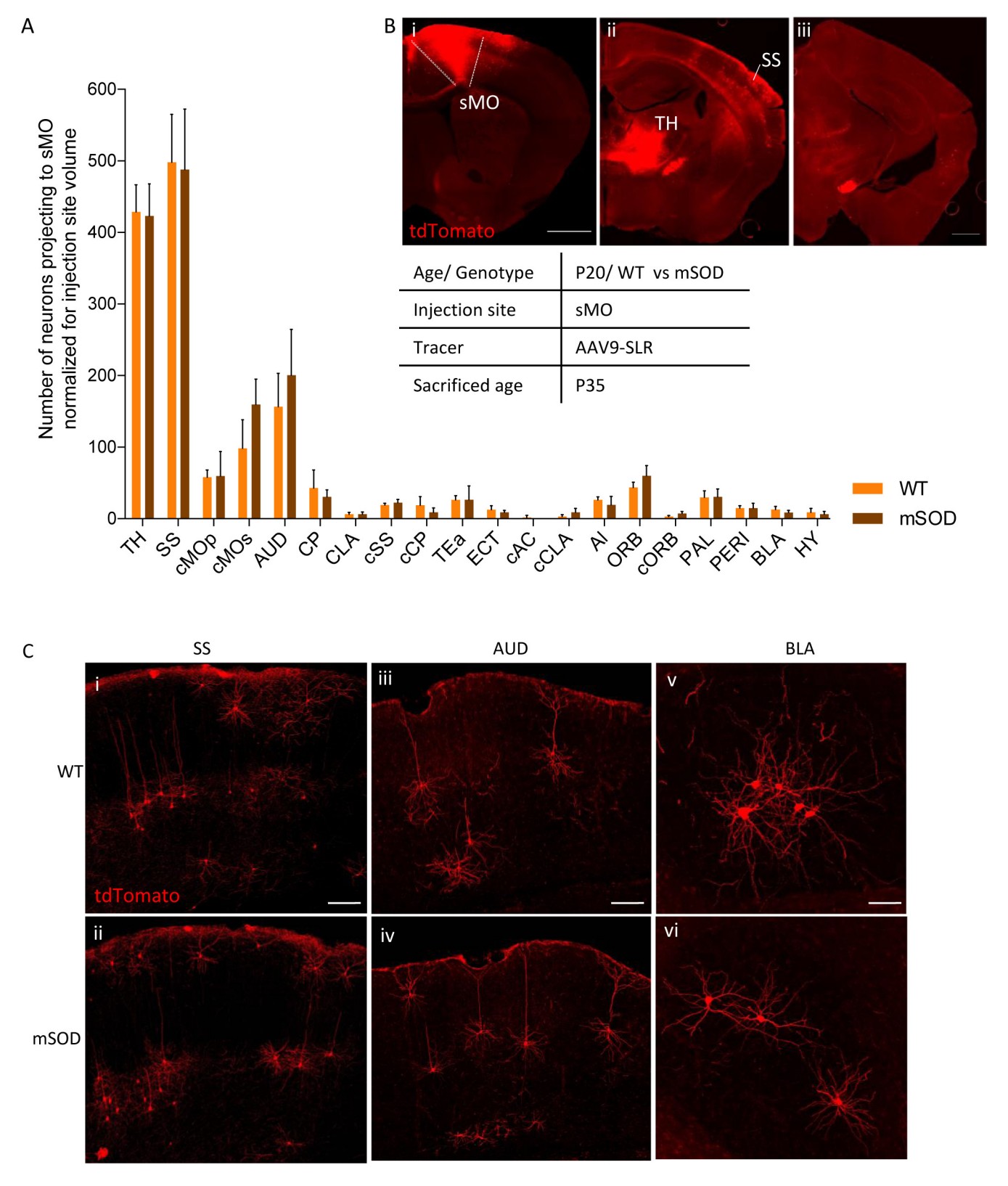

**Figure 5.** No change in projections to secondary motor cortex, at pre-symptomatic stage (P20). (**A**) Number of neurons, normalized for injection site volume, projecting to secondary motor cortex (sMO) in WT (N = 3) and mSOD (N = 3). No statistically significant differences in input to sMO were identified between mSOD and WT mice, although a trend toward increased projection in mSOD was found in contralateral sMO and AUD (two-way

*Figure 5 continued on next page*

*Figure 5 continued*

ANOVA with Sidak correction for multiple comparison). Detailed statistic in *Supplementary file 2k*. Total neuronal count has been reported in *Supplementary file 2j*. (B i-ii-iii) Representative brain sections displaying main regions targeting secondary motor cortex, scale bar 500 µm. (C) Representative images of neurons labeled by retrograde tracing from SS (i–ii), cAUD (iii-iv) and BLA (v–vi). Scale bar 150 µm. Analysis for total neuronal count is provided in *Supplementary file 2h*.

DOI: https://doi.org/10.7554/eLife.36892.017

The following source data is available for figure 5:

**Source data 1.** Detailed statistic concerning projecting neurons to secondary motor cortex (sMO) in WT vs mSOD animals (P20) traced via AAV9-SLR injection.

DOI: https://doi.org/10.7554/eLife.36892.018

Taken together, these findings revealed a stage- and structure-dependent remodeling of input to pMO, characterized by the progressive increase in the number of neurons projecting to the pMO, with cortical areas affected at earlier stages and subcortical areas only at later stages.

## Pyramidal layer V neurons across multiple areas of the motor subnetwork display simultaneous loss of spines on basal dendrites

Having demonstrated an increase in projection to pMO from specific cortical and subcortical structures with a progression-dependent pattern, we sought to determine if this phenotype could be correlated with pathogenic processes ongoing in the projecting neurons themselves. In order to verify if the neurons projecting to pMO already displayed signs of ALS-related pathogenic process, we elected to use dendritic spine density as readout. In fact, dendritic spine loss has been reported to be one of the earliest morphological signs of pathological involvement of several cortical and subcortical neuronal subpopulations in ALS (*Ozdinler et al., 2011*; *Fogarty et al., 2016*; *Fogarty et al., 2017*). Pyramidal neurons located in layers II/III and V were acquired from: SS (whose projection to pMO was increased in early stage of the disease), cMOp (increased projection starting from intermediate stage of the disease) and AUD (increased projection only at the later stages). We also included a sample of layer V and layer II/III neurons from ipsilateral pMO (known to be affected by the disease already at early stages; *Fogarty et al., 2017*).

At early pre-symptomatic stage (when increased projections from SS, but not from AUD, are already detectable), overall spine density was comparable in neurons in SS, cMO and AUD projecting to pMO from WT and mSOD mice (one-way ANOVA, $F_{(13,108)}=0.8$, p=0.6317; *Figure 7A* and *Supplementary file 3b*). Interestingly, layer-V neurons projecting to the pMO in SS, AUD and cMO displayed a decrease in spine density from P60 (*Figure 7B*), ($F_{(13,119)}=7.8$, p<0.0001); at the same time-point, a decrease in dendritic spine density in neurons from layer V of pMO itself became evident (p=0.0071; in agreement with what previously reported; *Ozdinler et al., 2011*; detailed numerical values are reported in *Supplementary file 3c*). Unlike layer V, we found no statistically significant change in the spine density in layer II/III at P60 in any of the cortical areas considered. At P90, a significant decrease in spine density affected neurons projecting to pMO from all structures under scrutiny either in layer II/III either in layer V (effect size between −1.6 and −1.8; $F_{(13,128)}=47.8$, p<0.0001, *Figure 7C* and *Supplementary file 3d*).

In conclusion, the loss of spines did not temporally correlate with the increased-projections phenotype, suggesting that the latter may not be interpreted as a consequence of pathology in projecting neurons (within the limits of the spine-loss readout). In addition, we show that spine loss affects not only vulnerable neurons in pMO, but rather neurons across the whole motor subnetwork.

## misfSOD expression is not related to network remodeling nor to spine pathology

Next, we investigated if the build-up of misfolded SOD1 (misfSOD1) in neurons projecting to pMO may be related to the increased-projection phenotype. MisfSOD1 has been reported to be a hallmark of mSOD-ALS pathobiochemistry (*Bosco et al., 2010*) and has been identified as the earliest disease marker in vulnerable spinal MN (*Saxena et al., 2013*). To this aim, mice were injected with fluorescently labeled CTb in pMO at P20 and sacrificed at P35, in order to label the neuronal subpopulations projecting to pMO; misfSOD1 was revealed by the B8H10 monoclonal antibody against

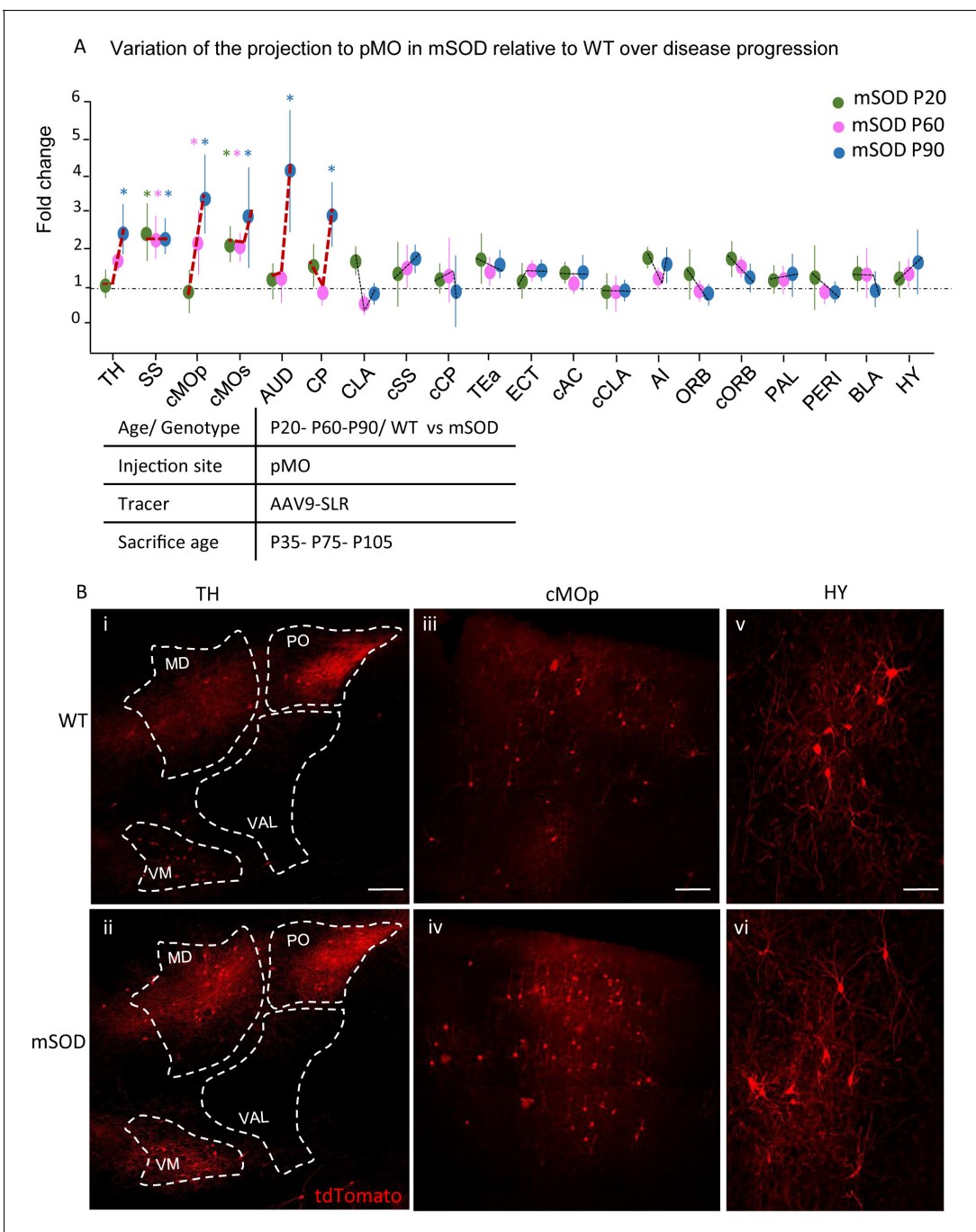

**Figure 6.** Progressive changes in projections to pMO in disease progression. (**A**) Projection neurons were mapped at P20, P60 and P90 timepoints; fold change of normalized total number of projecting neurons of mSOD vs WT was plotted for the three stages of disease progression. Values were expressed in terms of fold change over the corresponding WT for each time point. At P60 (WT N = 3, mSOD N = 4), increased connectivity persisted from SS (2.3 ± 0.5 in mSOD vs WT littermates, p=0.0108) and cMOs (2.0 ± 0.3 in mSOD vs WT littermates, p=0.0381), but, in addition, cMOp showed an increased projection to pMO (2.2 ± 0.8, p=0.0323 in mSOD vs WT littermates). At P90 (WT N = 5, mSOD N = 5), further significant increase in projection to pMO was found in: TH (2.4 ± 0.6 in mSOD vs WT littermates, p=0.0002), SS (2.2 ± 0.4 in mSOD vs WT littermates), cMOp (3.4 ± 1.0 in mSOD vs WT littermates, p<0.0001), cMOs (2.8 ± 1.3 in mSOD vs WT littermates, p<0.0001), AUD (4.0 ± 1.6 in mSOD vs WT littermates, p<0.0001) and CP (2.9 ± 0.7 in mSOD vs WT littermates, p<0.0001). (**B**) Representative images, for P90 stage, of neurons labeled by retrograde tracing from TH (**i–ii**), cMOp (**iii-iv**) and HY (**v–vi**). Scale bar 150 μm. Two-way ANOVA with Sidak correction for multiple comparison. Detailed statistic provided in *Supplementary file 3a*. Analysis of number of neurons in pMO for later stage (P90) mSOD animals provided in *Figure 6—figure supplement 1A–B*.
DOI: https://doi.org/10.7554/eLife.36892.019

The following source data and figure supplements are available for figure 6:

*Figure 6 continued on next page*

*Figure 6 continued*

**Source data 1.** Detailed statistic concerning projecting neurons to primary motor cortex (pMO) in WT vs mSOD animals, traced via AAV9-SLR injection, during disease progression.
DOI: https://doi.org/10.7554/eLife.36892.022
**Figure supplement 1.** No neuronal loss in pMO of mSOD animal (P90).
DOI: https://doi.org/10.7554/eLife.36892.020
**Figure supplement 1—source data 1.** Detailed statistic concerning the number of NeuN +neurons in the motor cortex of adult mSOD animals (P90).
DOI: https://doi.org/10.7554/eLife.36892.021

one conformational epitope of misfolded SOD (*Pickles et al., 2013*). In pMO, where upper MN are mainly (but not exclusively) localized, already at this age 41 ± 7% of neurons in pMO layer V were positive for misfSOD +compared to only 1 ± 0.5% of layer II/III neurons. MisfSOD1 burden (assessed by the immunofluorescence intenty) in misfSOD1 +neurons was comparable in pMO layer II/III (1798±355) and layer V (2189 ± 393; p>0.05). We then identified the neurons in SS and AUD projecting to pMO by CTb retrograde labelling (the non-viral approach was used to avoid possible artifacts due to the over-expression of ZSGreen) and assessed the mSOD burden by immunostaining (*Figure 8A–C*). Both SS and AUD included a misfSOD +subset of neurons in layer II/III (20 ± 5%) and layer V (54 ± 6%). However, when we considered the CTb + and CTb- subpopulations of neurons, we found that, in layers II/III and V the population of neurons projecting to pMO (CTb+) displayed a significantly lower burden of misfSOD1 than the overall neuronal population of the CTb- neurons (effect size between −0.9 and −1.4; one-way ANOVA, $F_{(11,1242)}$ = 35.3, p<0.0001; *Figure 8A–C* and detailed numerical values in *Supplementary file 3e*) in both cortical layers of SS (p<0.0001) as well as in both cortical layers of AUD (p<0.0001). In order to verify that this finding was not due to a confounding effect of CTb itself, we traced a different subpopulation of SS neurons, namely those involved in inter-hemispheric projections; we injected fluorescently labeled CTb in contralateral SS of mSOD1 mice at P30 and we assessed the burden of misf SOD in CTb + and CTb- neurons (*Figure 8D; E*). In contrast to what observed in the subpopulation connected to pMO, SS subpopulation projecting to contralateral SS displayed a range of misfSOD levels comparable to non-projecting, nearby neurons (p=0.2696) including, in both cases, neurons with high burden and neurons with low burden of mSOD (*Figure 8D–E* and *Supplementary file 3f*). Thus, taken together, these findings suggest that although within regions part of the motor subnetwork there is a mixed population of neurons with high and low burden of misfSOD, the neurons directly projecting to pMO constitute a subpopulation with comparatively low misfSOD burden.

Next, we asked if the spine loss phenotype observed in neurons projecting to pMO at later timepoints could be correlated with misfSOD build-up. To this aim, P60 mSOD/ZsGreen mice were injected with rAA9-SLR in pMO (N = 3) and the burden of misfSOD was assessed in SS ZsGreen + neurons (layer II/III and layer V) by immunostaining with the B8H10 monoclonal antibody (*Figure 8F; G*). Based on misfSOD intensity thresholds (the 10th and 90th percentile values of the single-neurons distribution shown in *Figure 8B*), we discriminated two population of ZsGreen + neurons (high and low misfSOD expression, either in layer II/III either in layer V) for which the spine density on basal dendrites was counted. Interestingly, despite the extreme difference in misfSOD burden, spine density was comparable in two groups and was within the range of WT mice for layer II/III neurons and reduced (compared to WT) for layer V neurons (in agreement with the values in *Figure 7B*; statistic in *Supplementary file 3g*), suggesting that spine loss may be a non-cell-autonomous phenomenon not directly related to the burden of mistSOD.

### Pattern of functional connectivity alterations in ALS patients displays similarity to the changes in projections in ALS mouse model

Having identified a subgroup of cortical areas and subcortical structures whose projection to pMO was increased in mSOD mice, we set out to verify if a similar set of areas display increased connectivity to pMO in ALs patients. To this aim, we analyzed the connectivity pattern in a large cohort of 71 ALS patients compared to 28 healthy controls by using 'resting-state' fMRI. The connectivity patterns were studied using a connectome-based approach in order to investigate region-to-region functional connectivity in a manner allowing comparison of results between the patient and mSOD mouse studies. Each of the a priori defined seed regions allowed for the identification of well-known functional

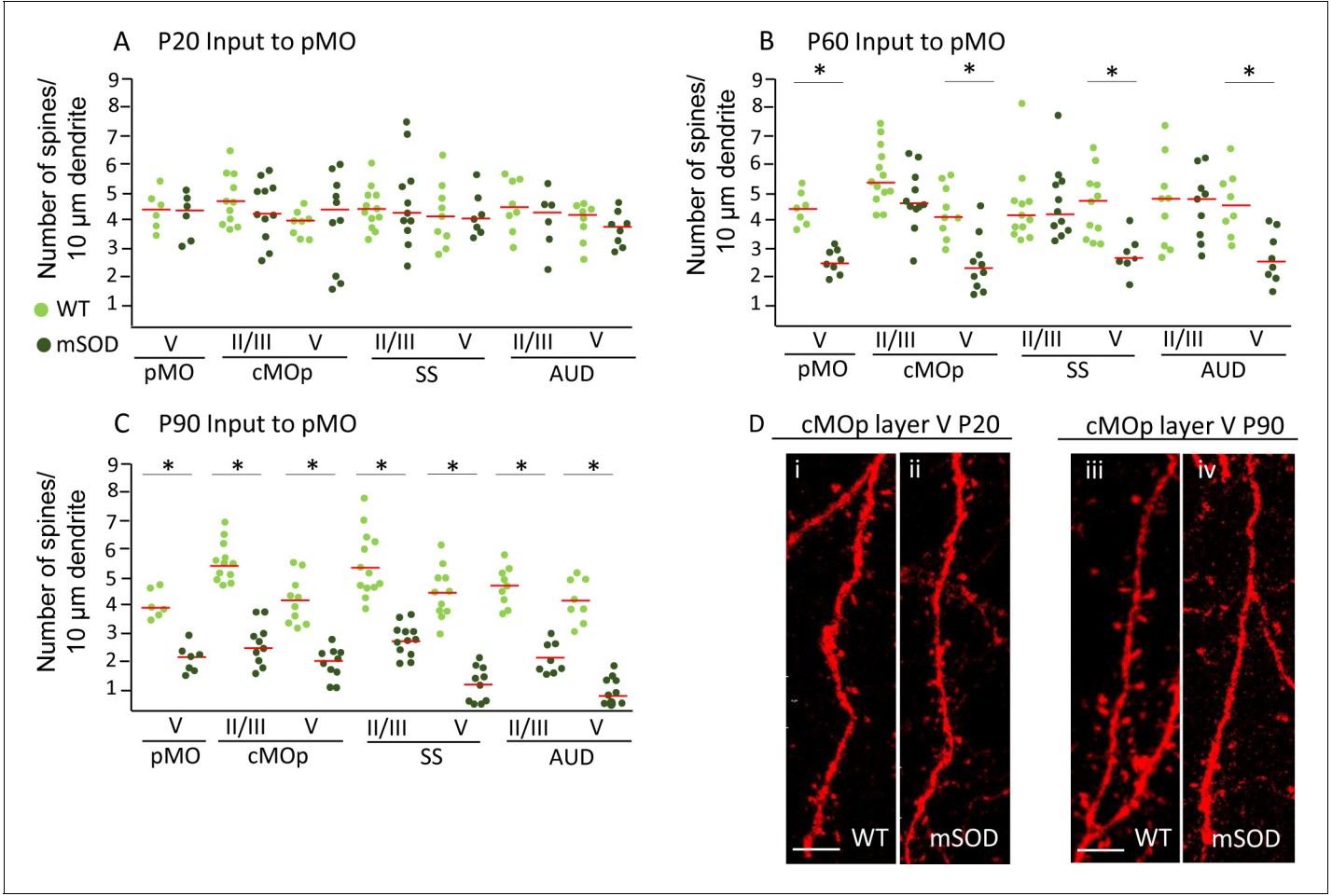

**Figure 7.** Pyramidal neurons projecting to pMO display simultaneous spines loss on basal dendrites during disease progression independently of misfolded SOD accumulation. (A) Spine density on pyramidal neurons projecting to primary at P20; each dot representing average spine density for one single neuron. Spine density was comparable in pyramidal neurons projecting to pMO in WT and mSOD mice. Detailed statistic provided in *Supplementary file 3b*. (B) Neurons projecting to primary motor cortex started to display spine loss at P60. Pyramidal neurons in layer V belonging to cMOp (4 ± 1 spines/10 µm in WT vs 2 ± 1 spines/10 µm in mSOD; p=0.0030), SS (4 ± 1 spines/10 µm in WT vs 3 ± 1 spines/10 µm in mSOD; p=0.0047) and AUD (5 ± 1 spines/10 µm in WT vs 3 ± 1 spines/10 µm in mSOD; p=0.0034) of mSOD mice displayed a significant decrease in spine density compared to their WT counterpart animals. Projecting neurons in layer II/III displayed a spine density comparable to WT. Detailed statistic provided in *Supplementary file 3c*. (C) Significant decrease in spine density affected the whole cortex when the latest stage was reached, at age of P90: cMOp layer II/III (WT 6 ± 1 spines/10 µm in WT vs 2 ± 1 spines/10 µm in mSOD, p=0.0001), layer V (4 ± 1 spines/10 µm in WT vs 2 ± 0.5 spines/10 µm in mSOD, p<0.0001); SS layer II/III (6 ± 1 spines/10 µm in WT vs 3 ± 1 spines/10 µm in mSOD, p<0.0001), layer V (4 ± 0.6 spines/10 µm in WT vs 1 ± 0.6 spines/10 µm in mSOD, p<0.0001); AUD layer II/III (5 ± 1 spines/10 µm in WT vs 2 ± 0.6 spines/10 µm in mSOD, p<0.0001), layer V (4 ± 1 spines/10 µm in WT vs 1 ± 0.5 spines/10 µm in mSOD, p<0.0001). Detailed statistic in *Supplementary file 3d*. (D i-ii) comparison of basal dendrite stretched between WT and mSOD at P20 in contralateral pMO layer V, scale bar 5 µm. **D iii-iv**: comparison of basal dendrite stretched between WT and mSOD at P90 in contralateral pMO layer V. Scale bar 5 µm. One-way ANOVA with Sidak correction.

DOI: https://doi.org/10.7554/eLife.36892.023
The following source data is available for figure 7:

**Source data 1.** Detailed statistic concerning spine density on pyramidal neurons projecting to primary motor cortex during disease progression (P20, P60, P90).
DOI: https://doi.org/10.7554/eLife.36892.024

brain networks in healthy humans (*Smith et al., 2009*) (*Figure 9A*), which indicated robust seed locations for the functional connectome analysis. Overall functional connectivity analysis demonstrated no statistically significant differences between groups (t = −0.51, p=0.613), which rules out a potential bias in the overall level of connectivity (*van den Heuvel et al., 2017*). However, network-based statistics of functional connectome data revealed significantly increased functional connectivity

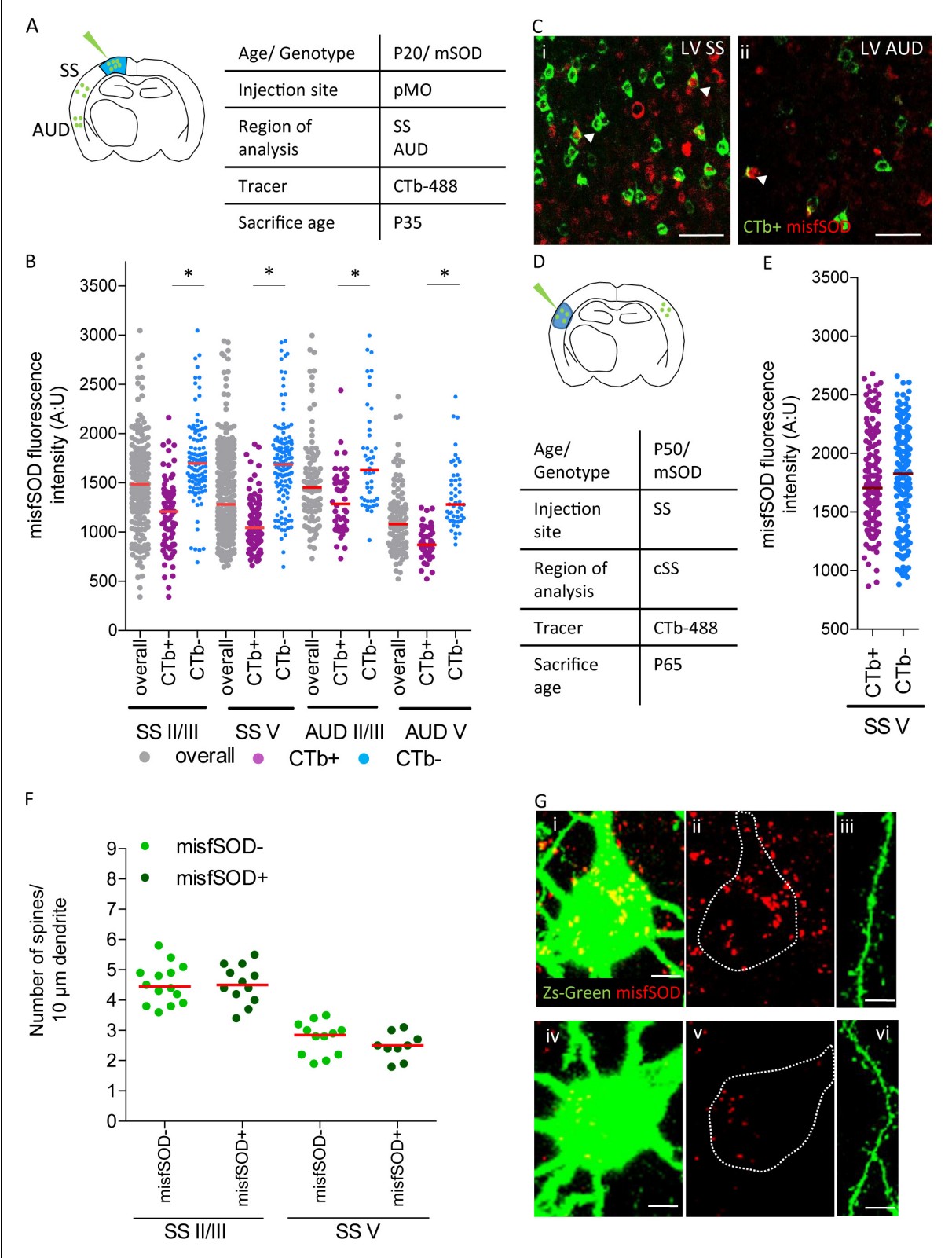

**Figure 8.** misfSOD expression does not trigger network remodeling nor loss of spines in basal dendrites of connected pyramidal neurons. (**A**) Experimental design depicting injection site in pMO via choleratoxin (CTb) with two regions that have been investigated for misfSOD immunostaining intensity (SS and AUD). (**B**) Comparison of misfSOD expression between neurons CTb + and CTb- in layer II/III and layer V of SS and AUD in mSOD. In both areas, a subset of neurons in layer II/III and layer V displayed misfSOD accumulation at P20. Compared to the overall population of neurons in

*Figure 8 continued on next page*

*Figure 8 continued*

layer II/III and V (CTb-), the populations of neurons projecting to pMO (CTb+) displayed a lower burden of misfSOD: SS layer II/III CTb + 1179 ± 350, CTb- 1730±418, p<0.0001; SS layer V CTb + 1073 ± 238, CTb- 1706±445, p<0.0001; AUD II/III CTb + 1318 ± 325, CTb- 1755±518, p<0.0001; AUD layer V CTb + 888 ± 164, CTb- 1372 ± 329, p<0.0001. One-way ANOVA with Sidak correction, detailed statistic reported in *Supplementary file 3e*. (**C i-ii**) Magnification of SS and AUD layer V respectively, red stains misfSOD antibody, green stains for CTb+, scale bar 50 µm. (**D**) Experimental design depicting injection site in SS via CTb with contralateral SS investigated for misfSOD immunostaining intensity. (**E**) comparison of misfSOD expression between neurons CTb + and CTb- in layer V of contralateral SS in mSOD. Injection of CTb was performed in SS. The populations of neurons projecting to SS display the same burden of misfSOD: CTb + 1770 ± 400, CTb- 1782±442, (t Test p=0.2696). Detailed statistic is reported in *Supplementary file 3f*. (**F**) Spine density analysis on two differential neuronal populations, misfSOD- and misfSOD+, projecting from SS to primary motor cortex at P60; each dot representing average spine density for one single neuron. No difference has been detected (one-way ANOVA with Sidak correction, SS layer II/III p>0.9999, SS layer V p>0.9999), detailed statistic reported in *Supplementary file 3g*. (**G**) Representative images of misfSOD – and misfSOD + neuron in SS layer V. i-ii-iii: misfSOD+ neuron displaying ZsGreen (green) and misfSOD (red, scale bar 20 µm) together with a stretch of its basal dendrite (scale bar 5 µm). iv-v-vi: misfSOD- neuron displaying ZsGreen (green) and misfSOD (red, scale bar 20 µm) together with a stretch of its basal dendrite (scale bar 5 µm).

DOI: https://doi.org/10.7554/eLife.36892.025

The following source data is available for figure 8:

**Source data 1.** Detailed statistic concerning comparison of misfSOD expression between neurons CTb + and CTb- in layer II/III and layer V of SS and AUD in mSOD.

DOI: https://doi.org/10.7554/eLife.36892.026

(hyper-connectivity) between regions frontal to the pMO, namely the dorsolateral prefrontal association cortex and the pMO itself, both homolaterally and contralaterally (t ≥ 2.81, p≤0.0065, corrected; *Figure 9B*). Functional connectivity between left and right primary somatosensory cortex was decreased (t = 2.93, p=0.0048) and no effect was detected between thalamic nuclei and pMO or in visual cortex. Thus, the fMRI data from ALS patients confirmed the occurrence of a large-scale remodeling in the motor subnetwork (as observed in the murine dataset and confirming previously published data; *Schulthess et al., 2016*) and showed a predominant increase in connectivity

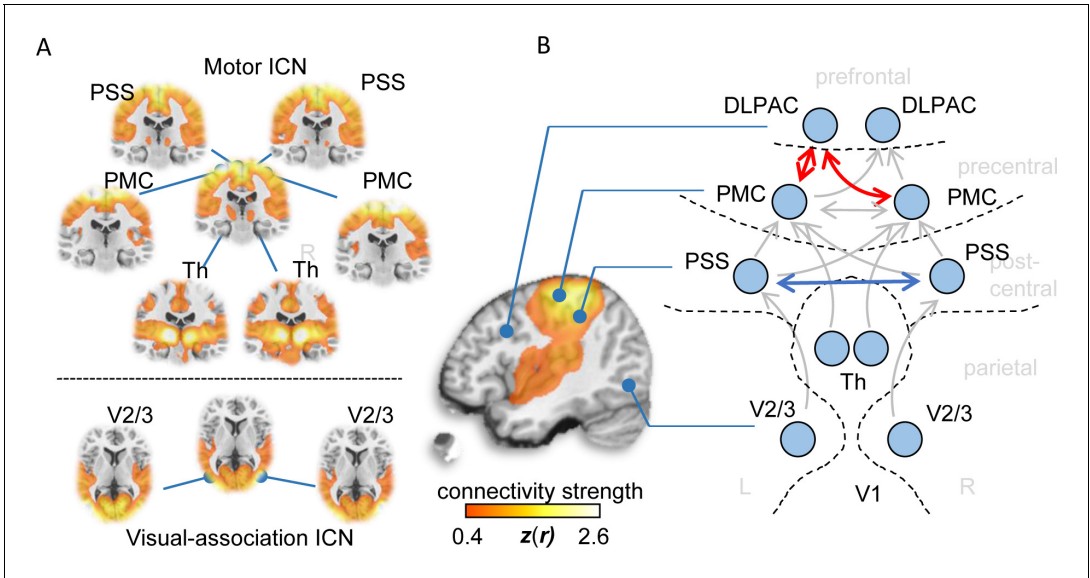

**Figure 9.** Functional connectivity alterations in human ALS. (**A**) BOLD synchronization illustrated as coronal (upper panel) and axial connectivity (lower) heat maps showing voxel-wise Fisher's *r*-to-*z* transformed correlation coefficients (thresholded for |z(r)|≥0.4) for which the fMRI BOLD signal was correlated with the respective seed-voxel forming the motor system (upper panel) and the visual association system (lower panel) in healthy elderly human subjects (N = 28). (**B**) Pairwise region-to-region functional connectivity strength analysis between schematically illustrated regions (blue circles) corresponding to seed voxels (as shown in A) revealed significantly altered functional motor system connectivity (red arrows) in ALS patients (N = 71) compared to matched healthy controls (N = 28). PMC = Primary motor cortex, PSS = Primary somatosensory cortex, DLPAC = Dorsolateral prefrontal association cortex, V2/3 = Visual association area V2/V3, Th = thalamus.

DOI: https://doi.org/10.7554/eLife.36892.027

between pMO and both homolateral and contralateral frontal-lobe areas, showing similarity with the increase projection from contralateral rostral structures, such as sMO, observed in the mouse model.

## Discussion

In the present work, we have demonstrated, using viral-tracing approaches, that the motor subnetwork in an ALS mouse model undergoes distinct remodeling during disease progression with stage and structure-dependent effects: projection from SS is increased already in presymptomatic mice and remains stable, whereas projections from AUD and cMOs and cMOp either appear or increase further in the symptomatic stage. We show that this pattern displays similarities with the network remodeling in the form of increase functional connectivity changes observed by fMRI in ALS patients. Thus, we provide cellular-resolution data that contribute to analysis of large-scale architecture of motor circuits in ALS mouse model, which may provide entry points to understand changes occurring in human patients.

In order to map the input to the MO in mSOD and WT mice, a retrograde viral-tracing strategy was considered. We have introduced a new variant of AAV9 which displays significantly higher capacity for retrograde infection of neurons in vivo. Although retrograde tracing can be achieved with non-viral approaches, these methods suffer from limited sensitivity, stability and do not provide genetic access to the target neurons (*Tervo et al., 2016*). Retrograde tracing has been previously achieved in vivo using rabies-derived vectors (*Wickersham et al., 2007*), herpes simplex virus (*Ugolini et al., 1987*) or canine adenovirus (*Soudais et al., 2001*). These vectors, however, have been plagued by the intrinsic toxicity of the expressed viral proteins (as in the case of rabies - *Ginger et al., 2013* - although new rabies variants may have overcome this issue, see *Ciabatti et al., 2017*) or by the tendency to activate inflammation (*Muruve, 2004*). Recently, the development of retrograde AAV vectors have been undertaken to allow retrograde tracing with genetic access and low toxicity; in particular, a retrograde AAV2 variant has been reported (rAAV2; *Tervo et al., 2016*). Our AAV9 variant extends and complements the existing toolset of AAV vectors endowed with retrograde infectivity. Although rAAV2 belongs to the AAV2 serotype and may strongly bind heparan sulphate, our AAV9-derived retrograde AAV variant has the advantage of reduced binding to heparin sulfate (*Shen et al., 2011*) and therefore it is less affected by changes in brain extracellular matrix (e.g. due to neuroinflammation associated with neurodegeneration); AAV9 is only infrequently targeted by pre-existing immunity to AAVs (*Zhang et al., 2011*; *Saraiva et al., 2016*) and may provide the entry point to enhanced vectors for circuit-specific gene therapy in humans since AAV9 has been already successfully tested for this application (*Boutin et al., 2010*).

We first validated the tool by mapping the input connectivity to pMO and sMO at cellular resolution; notably, the discreet pattern of regions containing retrogradely labelled neurons separated by others containing or no labeled cell, in a reproducible pattern and matching the list of brain structures projecting to pMO (e.g. *Hooks et al., 2013*) and the Allen Brain Connectivity atlas) suggest that the infection of distant neurons is not an artifact of passive viral diffusion or infection via blood supply (which would have resulted in a continuous or erratic distribution of labelled neurons). We further validated the reliability of our approach in the context of the mSOD1 mouse, verifying that viral-tracing results on SS and pMO could be reproduced by non-viral tracing (CTb) and that no abnormalities could be found in the projections to visual cortex, an area spared the pathogenic process. Nevertheless, AAV vectors relie on retrograde transport to reach the cell body (where expression of the genome takes place; *Castle et al., 2014b*) in process dependent on dynein/dynactin (*Castle et al., 2014a*) and one may speculate that the transport slow-down and dysfunction reported in ALS (*Bilsland et al., 2010*; *Marinkovic et al., 2012*) may bias the quantitative evaluation of the tracing experiments. However, the slowing of the transport would result in the decrease or delayed arrival of the AAV to the neuronal cell body, actually decreasing the chances of Cre expression and possibly resulting in decreased number of projecting neurons; therefore, this mechanism is unlikely to underlie the observed increase in the number of projecting neurons.

We exploited this new retrograde AAV9 to investigate in ALS models the qualitative and quantitative changes in the structure of the motor subnetwork, in particular of the projections to the pMO. A major finding of the present work is the increase in the numbers of neurons projecting from several cortical and subcortical regions to pMO; the increase is selective for some cortical areas only and the number of regions involved increases over time. The increase in the number of neurons

projecting to a fixed, small volume of pMO is most likely to be interpreted as due to the expansion of the axonal arborizations of these neurons in pMO as disease progresses and it is in agreement with the increased frequency of EPSPs observed early in disease course (*Fogarty et al., 2015*). What may be the driving force of this expansion? One may hypothesize that a pathogenic process ongoing in projecting neurons may drive the expansion; however, neither the burden of misfolded SOD1 nor the signs of structural disturbances (i.e. the decrease in spine density) were detectable in the areas displaying increased projection to pMO. On the other hand, it has been demonstrated in several models (e.g. in the neuromuscular unit), that silencing of either the pre- or post-synaptic site is sufficient to induce axonal sprouting (*Caroni et al., 1994*; *Caroni, 1997*). Since reduced firing of vulnerable spinal MN has been shown to precede degeneration (*Martínez-Silva et al., 2018*; *Roselli and Caroni, 2015*), it is tempting to speculate that the expansion of the projections may be driven by abnormal activity of vulnerable post-synaptic neurons in the pMO (including vulnerable corticospinal MN; *Ozdinler et al., 2011*). Indeed, in mSOD mice excitability changes appear to affect a larger set of neurons in pMO than previously thought (*Kim et al., 2017*).

When the progressive increase in projections to pMO detected in the tracing experiments are compared with the changes in connectivity reported here and by others (*Agosta et al., 2018*; *Menke et al., 2016*; *Schulthess et al., 2016*), several similarities emerge: (i) both in human and mouse data, we observe a progressive remodeling of the motor subnetwork, (ii) the remodeling of the motor subnetwork is detectable before the appearance of overt clinical or behavioral abnormalities, (iii) the remodeling of the motor subnetwork involves the primary motor cortex and more rostral, homolateral or contralateral structures (in the frontal lobe, prefrontal cortex or secondary motor cortex in human and mouse, respectively) and projecting to pMO, and (iv) both in human and mouse data the primary somatosensory cortex appears involved in the motor subnetwork remodeling (either in terms of increased projection to pMO in the mouse, or in terms of connectivity between left and right SS, SS, sensory-motor or SS to other cortical areas; *Douaud et al., 2011*; *Agosta et al., 2011*). Nevertheless, thalamocortical connectivity does not appear to be altered in the functional connectivity studies.

Based on these similarities, might the increase in projection to pMO be a leading mechanism underlying the functional hyperconnectivity in ALS patients? Although the relationship between structural and functional connectivity is not precise, severals studies have demonstrated that, at large-scale level, structural connectivity is a strong predictor of functional connectivity (*Honey et al., 2009*; *Honey et al., 2007*; *Damoiseaux and Greicius, 2009*; *Hermundstad et al., 2013*) and maps of functional and structural connectivity display high degrees of similarity (*van den Heuvel et al., 2009*). Moreover, studies in non-human primates comparing functional connectivity with anatomical tracing have revealed that anatomical connectivity strongly (although not exclusively) shape the functional connectivity (*Shen et al., 2012*; *Shen et al., 2015*). On the other hand, deducing structural connectivity from functional connectivity data can be fraught with inaccuracies, since indirect connections may contribute to it (*Adachi et al., 2013*). Thus, on the assumption that the observations made on the remodelling of the motor subnetwork in mouse models could be extrapolated to humans, the expansion of projections to pMO may contribute a structural substrate of the functional hyperconnectivity observed in fMRI studies. Notably, alternative models have been proposed to explain the hyperconnectivity phenotype. Changes in functional connectivity could also result from an alteration in the local excitation/inhibition balance due to the dysfunction of inhibitory interneurons (*Douaud et al., 2011*): in fact, pathological studies have found that calbindin-positive GABAergic interneurons undergo degeneration in ALS (*Maekawa et al., 2004*), lending support to the hypothesis that loss of inhibitory tone contributes to clinical dysfunction in ALS (*Turner and Kiernan, 2012*). The two models are actually not mutually exclusive: the pattern of either increased either decreased connectivity in resting-state (*Jelsone-Swain et al., 2010*; *Mohammadi et al., 2009*; *Zhou et al., 2013*; *Agosta et al., 2013*; *Douaud et al., 2011*) and task-based fMRI (*Heimrath et al., 2014*; *Konrad et al., 2002*; *Lulé et al., 2009*) may be the result of ongoing degeneration and loss of axons (*Dadon-Nachum et al., 2011*; *Sarica et al., 2014*) accompanied by potentially compensative phenomenon (*Bernier et al., 2017*). The observed increase in projection from cortical and subcortical areas might be an early adaptive response to ongoing cell dysfunction up to the point where a critical cell loss is reached and a disconnection syndrome begins to emerge (*Gorges et al., 2017*).

In addition to the study of large-scale patterns of projections to vulnerable cortical areas, the use of retrograde viral vectors has allowed the access to morphological features of a distinct population of

neurons selected by connectivity rules (i.e. all neurons projecting to pMO). Our analysis has revealed a stage-dependent loss of spines in all neurons projecting to pMO (first in layer V, then in layer II/III), which appears independent of misfSOD burden and involves neuronal populations in areas whose vulnerability was previously unrecognized, such as AUD neurons. Spine loss has been detected as a morphological counterpart of the early stages of the degenerative process ongoing in the layer V of the motor cortex of mSOD mice (*Fogarty et al., 2015*). The longitudinal study of multiple brain areas in Golgi-staining preparations (*Fogarty et al., 2016*) has revealed the early and progressive spine loss affecting the basal dendrites of pyramidal neurons from MO, medial pre-frontal (MPFC) cortex; notably, SS neurons appeared to be affected only very late in disease progression (P120; *Fogarty et al., 2016*). By identifying a neuronal subpopulation of SS based on connectivity rules (i.e. only the neurons projecting to pMO), we report an earlier loss of spines than previously though (P50 vs P120). This finding not only underscores the importance of assessing disease manifestations in homogeneous (and characterized by connectivity rules) populations but also suggests that spine loss may be a phenomenon occurring in a larger set of neurons, possibly across the whole motor subnetwork.

Our study displays some limitations. Firstly, in order to maintain tractability while still addressing the core question of the large-scale remodeling of the structure of projections in the motor subnetwork, we decided to limit the analysis to the forebrain, avoiding the analysis of brainstem nuclei and cerebellum. Although these regions provide relevant input to the motor subsystem, they are not usually identified in fMRI studies of ALS and therefore their contribution to the definition of the hyperconnectivity phenotype may be more limited. Second, although it is desirable to have large number of biological replicates, the need to maintain the genotype blinding and to manually identify and anatomically annotate each neuron (more than 150,000 for the present work) did not allow for a n > 3–6, thus underpowering the study for the comparison of structures with fewer number of projecting neurons. Although the present study is in line with the numerosity currently manageable in manual reconstructions (*Hooks et al., 2013*; *Yamawaki et al., 2016*; *DeNardo et al., 2015*; *Tervo et al., 2016*; *Kohl et al., 2018*), the advancement of machine learning and automated neuronal annotation (*Fürth et al., 2018*) may allow a more comprehensive quantitative analysis of input network to pMO.

Third, we did not consider ipsilateral sMO and pMO for cell counting in our analysis. Due to the proximity of pMO and sMO, we could not have conclusively excluded the possibility that infected neurons in the ipsilateral sMO were derived from retrograde infection or virus diffusion, introducing ambiguity to the connectivity map. Further refinement of the retrograde-tracing approach is ongoing to provide a whole brain, full resolution map of the connectivity of the motor subsystem in ALS.

Taken together, our circuit-tracing data in mice and imaging data in humans converge to show that a significant and progressive remodeling of cortical circuits takes places in ALS and that this phenotype may be amenable to investigation in animal models. The mechanistic drivers of this remodeling are currently unclear, although it is possible to speculate that homeostatic mechanisms may be activated by alterations in excitability of pMO neurons, including corticospinal motor neurons (*Kim et al., 2017*). In fact, pathological decrease or increase in the firing properties of vulnerable neuronal subpopulations appears to be a shared phenotype of multiple neurodegenerative conditions (*Roselli and Caroni, 2015*) and may be one of the driving forces of adaptive cortical remodeling in neurodegeneration (*LANDSCAPE Consortium et al., 2015*; *Rosskopf et al., 2017*; *Rosskopf et al., 2018*). Thus, the combination of the analysis of large-scale projection patterns (enabled by retrograde viral tracing) with chronic manipulation of excitability and firing (such as chemiogenetics; *Saxena et al., 2013*) may prove fundamental to link functional and structural cortical phenotypes.

## Materials and methods

**Key resources table**

| Reagent type (species) or resource | Designation | Source of reference | Identifiers/RRIDS | Additional information |
|---|---|---|---|---|
| Antibody | Anti-Red Fluorescece Protein (RFP) | Rockland | 600-401-379/ RRID: AB_828390 | (1:1000) |

*Continued on next page*

*Continued*

| Reagent type (species) or resource | Designation | Source of reference | Identifiers/RRIDS | Additional information |
|---|---|---|---|---|
| Antibody | DAPI | Invitrogen | D1306 | (1:1000) |
| Antibody | Anti-Tyrosin-Hydroxilase | Sigma | T2928/ RRID: AB_477569 | (1:4000) |
| Antibody | Anti-misfolded SOD1 (B8H10) | MediMabs | MM-0070/ RRID:AB_10015296 | (1:1000) |
| Antibody | Donkey anti rabbit Alexa-568 | Life Technologies | A10042 | (1:500) |
| Antibody | Anti NeuN | Millipore | MAB377/ RRID:AB_2298772 | (1:100) |
| Antibody | Donkey anti-mouse Alexa-488 | Life Technologies | A21202 | (1:500) |
| Strain, strain background (*Mus musculus*) | B6SJL-Tg(SOD1 *G93A)1Gur/J | Jackson Laboratories | RRID:IMSR_JAX:002726 | high-copy, henceforth mSOD |
| Strain, strain background (*Mus musculus*) | B6.Cg-Gt(ROSA) 26Sortm6/ (CAG-ZsGreen)Hze/J | Jackson Laboratories | 007906 | henceforth ZsGreen-ROSA26 |
| Strain, strain background (*Mus musculus*) | B6.Cg-Gt(ROSA) 26Sortm6/(CAG-Td Tomato)Hze/J | Jackson Laboratories | 007914 | henceforth tdTomato-ROSA26 |
| Chemical compound, drug | Phenol Red Solution | Sigma-Aldrich | P0290 | |
| Chemical compound, drug | Protease inhibitor mix | Serva | 39101.03 | |
| Chemical compound, drug | Polyethylenimine (PEI) | Polysciences | 23966 | |
| Chemical compound, drug | Optiprep | Progen | 1114542 | |
| Chemical compound, drug | Buprenorphine | Reckitt Benckiser | | |
| Chemical compound, drug | Meloxicam | Böhringer Ingelheim | | |
| Chemical compound, drug | Ketamine 10% | WDT | | |
| Chemical compound, drug | Rompun 2% (Xylazin) | Bayer | | |
| Recombinant DNA reagent | AAV9-RGDLRVS -CMV-Cre | *Varadi et al., 2012* | | |
| Recombinant DNA reagent | AAV9-SLRSPPS -CMV-Cre | *Varadi et al., 2012* | | |
| Recombinant DNA reagent | AAV9-NSSRFTP -CMV-Cre | *Varadi et al., 2012* | | |
| Recombinant DNA reagent | WT-AAV2- CMV-Cre | *Werfel et al., 2014* | | |
| Recombinant DNA reagent | WT-AAV9-CMV-Cre | *Werfel et al., 2014* | | |

| Reagent type (species) or resource | Designation | Source of reference | Identifiers/RRIDS | Additional information |
|---|---|---|---|---|
| Peptide, recombinant protein | Alexa-488-conjugated cholera toxin B | Invitrogen | C34775 | |
| Peptide, recombinant protein | Alexa-647-conjugated cholera toxin B | Invitrogen | C34778 | |

## Animals

All animal experiments in this study were performed in agreement with the guidelines for the welfare of experimental animals issued by the Federal Government of Germany and by the local ethics committee (Ulm University) and authorized under licence no. 1312. The following strains of transgenic mice were obtained from Jackson Laboratories: B6SJL73 Tg(SOD1*G93A)1Gur/J (high-copy, henceforth mSOD), B6.Cg-Gt(ROSA)26Sortm6/(CAG74 ZsGreen)Hze/J (henceforth ZsGreen-ROSA26) and B6.Cg-Gt(ROSA)26Sortm6/(CAG75 TdTomato)Hze/J (henceforth tdTomato-ROSA26). For the generation of double-transgenic lines, heterozygous male mice from the mSOD line were crossed with females homozygous for the reporter cassette (either ZsGreen or tdTomato lines). Resulting progeny was 50% WT/ROSA26$^{ZsGreen}$+ and 50% mSOD/ROSA26$^{ZsGreen}$+. Under the repression of the STOP cassette in the ROSA26 locus, the fluorescent protein (either ZsGreen either tdTomato) is not expressed in basal conditions in the reporter lines; however, upon expression of the Cre recombinase (in our experimental design, via an AAV vector) leads to the excision of the STOP cassette (since it is flanked by the lox recombination sites) and results in the expression of the reporter (*Gerfen et al., 2013*).

Male mice were used for experimental purposes at post-natal day (P)20, P60 and P90 (sacrificed at P35, P75 and P105). Mice were housed at 3–4 animals per cage, with unlimited access to food and water, a light/dark cycle of 14/10 hr and humidity between 40% and 60%. All mice expressing mSOD were routinely tested for motor impairment and euthanized in case of overt motor disability. All male mice from a given litter were employed and the experimenter was kept blind to their genotype: injection, perfusion, brain section, imaging and neuronal counts and anatomical annotation; the genotype was revealed only when each mouse was assigned to a group for the statistical analysis.

All efforts were made to comply to the '3R' guidelines. In order to maintain the data analysis tractable and in absence of previous data on quantitative analysis of input connectivity in ALS mice, we estimated the sample size based on the mean number and standard deviation of thalamic neurons at P90, anticipating a genotype effect of at least 30–40%, with $\alpha$ = 0.05% and 90% power, resulting in group size between 3 and 5.

## Generation and production of AAVs

AAV-vectors were generated using a three plasmid standard protocol (*Jungmann et al., 2017*) with a pscAAV-CMV-Cre vector genome plasmid, derived from pscAAV-CMVEnh/MLC0.26-Cre (*Werfel et al., 2014*), pDGdeltaVP providing helper sequences, and a distinct rep-cap-helper plasmid with either AAV2 or AAV9 wild-type capsid genomes (resulting in WT-AAV2-CMV-Cre or WT-AAV9-CMV-Cre, respectively), or AAV9 variants containing distinct heptapeptide motifs reported previously (*Varadi et al., 2012*), resulting in AAV9-RGDLRVS-CMV-Cre, AAV9-SLRSPPS-CMV-Cre and AAV9-NSSRFTP-CMV-Cre. Vectors were purified and concentrated as described before (*Jungmann et al., 2017*).

## Intracerebral injections

Intracerebral injection of AAV vectors was performed as previously reported (*Karunakaran et al., 2016*). Briefly, mice were administered buprenorphine (0.01 mg/Kg; Reckitt Beckiser Healthcare, Berkshire, UK) and meloxicam (1.0 mg/Kg; Böhringer Ingelheim, Biberach an der Riß, Germany) 30 min before being put into a stereotaxic frame (Bilaney Consultants GmbH D-40211 Düsseldorf, Germany) under continuous isoflurane anesthesia (4% in O2 at 800 ml/min). Skin scalp was incised to

expose the underlying bone and a hand-held micro drill was used to drill the burr hole in the opportune location. Stereotactic coordinates were chosen for each target area based on The Mouse Brain in Stereotaxic Coordinates (as in *Oh et al., 2014*) and corresponded to primary motor cortex (x = + 1.5, y =+ 1.0, z = −0.5), secondary motor cortex (x =+ 0.7, y =+ 1.0, z=-0–5), striatum (x =+ 1.0, y =+ 1.5, z = −3.5), visual cortex (x = −3.0, y =+ 2.0, z = −0.6) and primary somatosensory cortex (x =+ −1.0, y = −1.0, z = 0.5). 500 nl of viral suspension, mixed with an equal volume of 1% Fast Green solution (100 mg of Fast Green in 10 mL of $H_2O$), was injected using a pulled-glass capillary connected to a Picospritzer microfluidic device. The Fast-Green visible dye, used to visually monitor the injection, is quickly washed away in PBS and does not interfere with the imaging procedures. Injection was performed with 10 msec pulses every 30 s over a span of 5 min. The capillary was kept in place for 10 more minutes before being withdrawn to prevent backflow of the virus. Scalp skin was then stitched with Prolene 7.0 surgical thread and the animals were transferred to single cages with facilitated access to water and food for recovery. Animals were monitored for eventual neurological impairment for the following 72 hr and were administered additional doses of buprenorphine as needed.

Each of the following AAV vectors was independently injected at the titre of 5–9*$10^{13}$/ml: wild-type (WT) AAV9-CMV-Cre (WT-AAV9), WT-AAV2-CMV-Cre (WT-AAV2), AAV9-RGDLRVS-CMV-Cre (AAV9-RGD), AAV9-SLRSPPS-CMV-Cre (AAV9-SLR) and AAV9-NSSRFTP-CMV-Cre (AAV9-NSS).

For cholera toxin injections, Alexa-488-conjugated or Alexa-647-conjugated cholera toxin B (CTb-488, Invitrogen) was diluted according to the manufacturer's instructions to 1 µg/µl by gentle mixing (no vortexing). 1.0 µl of CTb was injected using the same procedure as reported above.

## Immunohistochemistry

Mice were terminally anesthetized with cloralium hydrate and transcardially perfused with 50 ml ice-cold PBS followed by 2.5–3 ml/g of 4% paraformaldehyde (PFA) in phosphate buffered saline (PBS, pH 7.4). Brain samples were quickly dissected and post-fixed in 4% PFA in PBS at 4°C for 18 hr, washed in PBS and cryo-protected in 30% sucrose in PBS for 36 hr. Samples were then snap-frozen in OCT and sectioned at −15°C in a cryotome (Leica CM1950) to 70µm-thick sections serially collected. Free-floating brain sections were incubated for 2 hr at 24°C in blocking buffer (5%-donkey serum, 3% BSA, 0.3% Triton X-100), followed by incubation with primary antibodies, such as: rabbit anti-RFP (1:1000, Rockland Immunochemicals, Limerick, PA) for 24 hr at 4°C; mouse anti-Tyrosine Hydroxylase (1:2000, Sigma) for 24 hr at 4°C; mouse anti NeuN (1:100, Millipore) for 24 hr at 4°C; mouse monoclonal anti-misfolded SOD1 (B8H10, 1:1000, MediMabs) for 48 hr at 4°C. Brain sections were thereafter washed 3 × 45 min in PBS, 0.1% Triton, incubated for 2 hr at 24°C with the opportune secondary antibody combination in blocking buffer (donkey anti-rabbit Alexa-568; donkey anti-mouse Alexa-488, 1:500; Life Technologies), together with DAPI (0.1 µg/ml, Life Technologies), and washed again 3 × 45 min in PBS, 0.1% Triton. Brain sections were mounted using ProLong Gold Antifade (ThermoFisher) mounting medium.

## Image acquisition and analysis

In order to analyze the local infection rate of the virus in the dorsal striatum (DS), we identified the core of injection according to the brightness level. Four regions of interest (ROI, 300 × 300 µm² each) were drawn at its cardinal points and 500 µm³ of tissue was analyzed in total. For each ROI, reporter +neurons and DAPI +nuclei were counted. Local infectivity for each AAV variant was calculated by averaging ROI values for every mouse (N = 3 for each viral variant) and the values are expressed as a percentage of AAV9 variants over WT- AAV9. For analysis of the retrograde infectivity in motor cortex (MO), substantia nigra (SNc), thalamus (TH) and lateral geniculate nucleus (LGN), brain slices corresponding to our target brain regions were cut (N = 3 for each viral variant). Reporter +soma were counted for every region and the values are expressed in terms of percentage over WT-AAV9.

For analysis of input connectivity to primary motor cortex (pMO, considering all time points analyzed: WT N = 11, SOD N = 12) and visual cortex (VIS, WT N = 3, SOD N = 3) via AAV9-SLR variant, all sections corresponding to each brain were scanned using the BZ-9000 fluorescence microscope (KEYENCE) equipped with a 4x objective, with exposure time set at 200 msec. All images were acquired in a 12bit format. Images were processed using the ImageJ software suite: images of

individual brain sections were loaded, background was subtracted offline via the appropriate tool to unambiguously identify reporter +cells. Labeled cells were positively identified as neurons (in contrast to non-neuronal cells or to dense clusters of axons) by the detection of a round soma surrounded by basal dendrites and, in cortical neurons, by the presence of an apical dendritic shaft. The soma of fluorescently labeled neurons was manually identified and annotated for anatomical brain regions according to Allen Brain Atlas coordinates. Total numbers of neurons were normalized (*Oh et al., 2014*) by the volume of the injection site in pMO (AAV9- SLR: 1.3 ± 0.6 mm$^3$, N = 23) and VIS (AAV9-SLR: 1.1 ± 0.5 mm$^3$, N = 6). Cortical layers were discriminated according to positional criteria and using the tissue autofluorescence (which highlights nuclei in negative, since they are not strongly autofluorescent) and the region of interest (ROI, rectangular in shape, approximately 200 µm high) was placed approximately 100 µm (upper border) ventral to the cortical (pial) surface for layer II/III, whereas for layer V and VI the lower border of the ROI was placed 300 µm dorsal of, or in contact with, respectively, the boundary between gray and white matter. The correctness of this positions was checked in a subset of NeuN and DAPI-stained sections.

For analysis projecting neurons we injected a small volume (500 nl out of a suspension of a titre of 5–9*10$^{13}$/ml of virus suspension) of AAV9-SLR virus suspension in the pMO corresponding to the whisker area (according to *Zingg et al., 2014*) or in the neighboring secondary motor cortex of tdTomato-ROSA26 or ZsGreen-ROSA26 reporter mice (age P20). After sacrificing the animals 15 days post-injection, brains were serially sectioned to 70 µm sections, and each section was digitized for manual annotation. An average of 5000 + neurons were manually identified (N = 6). In order to make the analysis tractable, cerebellum and brainstem nuclei were not included in the present study. Moreover, although ipsilateral sMO is a well-known input to pMO, its analysis was not performed because of technical issues: its proximity to the injection site made it difficult to distinguish between retrograde infection and spreading of the virus.

For analysis of input connectivity to pMO via cholera toxin (CTb Alexa Fluor 488 conjugate, Thermofisher, WT N = 3, SOD N = 3), brain sections were imaged with a Zeiss 710, 20x objective (NA 0.8), 0.9 optical zoom at 12 bit depth. CTb + soma were manually counted in contralateral pMO (cMOp) and somatosensory cortex (SS) following anatomical coordinates according to the Allen Brain Atlas and neuronal counting was normalized to the injection site volume (1.2 ± 07 mm$^3$, N = 6).

For analysis of neuronal cells number in pMO, 70 µm brain slices were cut for WT (N = 2) and SOD (N = 2) mice at the age of P90. Slices were stained for NeuN (Millipore). Images were acquired with AF6000 fluorescence microscope (Leica) equipped with a 10x objective, with exposure time set at 200 msec. All images were acquired in a 12-bit format. Single optical section images of pMO were analyzed for NeuN +soma via Imaris 7.6.5 software (Bitplane AG, 'Spot' tool, followed by manual correction).

For spine density analysis, fluorescent images were acquired with a Zeiss 710 confocal microscope, with a 63x oil-immersion objective (NA 1.40) and 0.9 optical zoom at 12bit depth. Cortical neurons expressing the reporter protein, therefore part of the motor network, were acquired in their full dimension, that is dendrites did not exit the plane of the coronal brain slice before reaching the dendritic terminus. Criteria for selection of neurons were anatomical and morphological: selected neurons were located in the desired cortical layer, showing the distinctive pyramidal shape and displaying at least three uncorrupted basal dendrites. For each neuron, three to six, 30–100 µm long, intermediate dendritic segments (10 µm distant from the tip) were acquired for spine counts. After image collection, each segment was analyzed for spine density using the Imaris 7.6.5 software (Bitplane AG, 'filament tracer' tool followed by manual correction). Only protrusions with a clear connection of the head of the spine to the dendrite shaft were counted as spines; small processes were classified as a spine only if they were <3 µm long and <0.8 µm in cross- sectional diameter (*Harris, 1999*). Spine density was then expressed as the number of spines per 10 µm dendrite length (*Saba et al., 2016*; *Jara et al., 2012*; *Vinsant et al., 2013*). Number of mice N = 3 for both WT and SOD for each time point.

To analyze misfSOD1 expression distribution within the motor network, we performed pMO injection of mSOD animals at P20 (sacrificed at P35, N = 3) with CTb. Brain slices were stained with the mouse monoclonal anti misfolded SOD1 B8H10 antibody (MediMabs, Montreal, Canada). Imaging was performed with a Zeiss 710, 20x objective (NA 0.8), 0.9 optical zoom at 12bit depth and images were analyzed using ImageJ. For misfSOD1 +neurons assessment, an ROI of 165 mm$^2$ was drawn for each cortical layer, a threshold was set and only neurons above the threshold were considered

positive. For counting CTb + neurons in layer II/III, we selected a total range of 200 μm, 100 μm away from the Pia and for layer V, we considered a total range of 200 μm, 400 μm away from the grey matter. The perimeter of misfSOD +nuclei was manually drawn for both CTb + and CTb- in order to calculate and compare their mean grey value.

## Statistical analysis - mouse experiments

Statistical analysis was performed with the GraphPad Prism six software suite. For comparing normalized neuronal counts between viral variants, ordinary one-way ANOVA with Dunnett multiple comparison test was used. The same test was applied for analysis of input to V1 using AAV-SLR and to pMO using CTb. For pMO and sMO input connectivity via AAV9-SLR, including comparison between the different cortical layers, either raw and normalized neuronal counts was compared by two-way ANOVA with Sidak correction for multiple comparison. Data regarding NeuN +neuronal count in primary motor cortex and misfSOD expression in CTb-labeled neurons have been analyzed via unpaired t-Test with Mann-Whitney correction. For the characterization of the variability of the dataset and for the description of the distribution of the data points, coefficient of variation (standard deviation/mean) and Skewness coefficient have been calculated (and are provided in tabulated format along with the mean, median and standard deviation).

Skewness value provides a descriptor of the slope of data distribution. In a perfect normal distribution curve, data distribute symmetrically on both sides of the peak, and the value of the skewness is zero. In a non-gaussian distribution, skewness value is positive when data piles up on the peak's left side and negative when data piles up on the peak's right side (and the tail points left). Skewness value between −2 and +2 are accepted as indication of non-biased data distribution (*George and Mallery, 2010*).

The magnitude of the difference between WT and mSOD was described using the parameter known as the effect size ((mean mSOD-mean WT)/SD of either group; calculated according to *Sullivan and Feinn, 2012*). Unlike significance tests, effect size is independent of sample size (*Ferguson, 2009*).

Data are displayed as mean ± SD and values for median, skewness and effect size are provided in tables as supplementary file. Statistical significance was set at p<0.05 before multiple-comparisons correction: statistical significance is indicated by p<0.05 and the magnitude of the effect is highlighted by the effect size. Percentage changes are reported in relation to relative WT.

## Functional connectivity data validation in humans

### Ethical approval of human study

All subjects included in the human study provided written informed consent according to institutional guidelines; the consent includes the declaration of the understanding of the study design, the agreement to the participation to the study, to the publication of the results, and to the data protection and anonymization procedures (under the chaptes Einwilligungserklaerung', 'Probandeninformation', 'Darstellung der Experimente', 'Datenschutzerklärung'). The study was approved by the Ethics Committee of Ulm University, Ulm, Germany (reference #19/12) and was performed in accordance with the ethical standards laid down in the 1964 Declaration of Helsinki and its later amendments.

### Data acquisition in humans

'Resting-state' functional magnetic resonance imaging (rs-fMRI) data were acquired from 71 symptomatic patients with ALS, diagnosed according to the recently revised El Escorial diagnostic criteria (*for The WFN Research Group On ALS/MND et al., 2015*), and 28 matched healthy controls (see *Table 1* for clinical and demographic characteristics). Human whole-brain based echo planar images (EPI) were acquired for all subjects at a 3T Siemens Allegra MR scanner (Siemens Medical, Erlangen, Germany) using a blood oxygen level dependent (BOLD) sensitized rs-fMRI sequence (30 transversal slices, gap 1 mm, $3 \times 3 \times 4$ mm$^3$ voxels, $64 \times 64 \times 30$ matrix, FOV $192 \times 192 \times 149$ mm$^3$, TE 30 ms, TR2000ms, flip angle 90°, 300 volumes).

### Preprocessing of functional data

The human functional data analysis followed a standardized procedure (*Gorges et al., 2014*) and was performed using the *Tensor Imaging and Fiber Tracking* (*TIFT*) software package (*Müller et al.,*

**Table 1.** Subject demographics and clinical characterization.

| | Healthy controls | ALS, all, $T_0$ | *p*-value |
|---|---|---|---|
| Subjects (number) | 28 | 71 | NA |
| Gender (male:female) | 15:13 | 39:32 | 1.000[*] |
| Age (years) | 54.8 (±12.9), 22.4—75.7 | 58.4 (±13.7), 19.7—85.1 | 0.228[†] |
| Duration of disease (month) | NA | 19.2 (±17.6), 2.6—84.7 | NA |
| Age of onset (years) | NA | 56.8 (±13.9), 19.2—84.6 | NA |
| ALSFRS-R[‡] | NA | 40 (±5), 24—48 | NA |
| Rate of disease progression[§] (1/month) | NA | 0.8 (±1.2), 0.0—7.8 | NA |

Data are shown as mean (±std), min—max. All values were computed using the MATLAB (The Mathworks Inc, Natick, MA) based 'Statistics Toolbox'.

[*]Fisher's exact test refers to comparison between all ALS patients and healthy controls.

[†]Two-sample unpaired *t*-test assuming unequal variances refers to comparison between all ALS patients and healthy controls.

[‡]ALSFRS-R, revised ALS Functional Rating Scale (maximum score 48, falling with increasing physical impairment).

[§]Rate of disease progression computed as (48 - ALSFRS-R)/(disease duration)(**Menke et al., 2014**). NA, not applicable.

DOI: https://doi.org/10.7554/eLife.36892.028

*2007*). Preprocessing included (1) spatial upsampling to an 1 × 1×1 mm³ isogrid (matrix, 256 × 256×256), (2) motion correction using a rigid brain transformation (six degrees of freedom), (3) stereotaxic Montreal Neurological Institute (MNI) normalization (*Brett et al., 2002*) using a landmark-based deformation approach (*Müller and Kassubek, 2013*), (4) temporal demeaning and linear detrending, (5) temporal bandpass filter (0.01 < *f* < 0.08 Hz), and (6) spatial smoothing using a 7 mm 3-dimensional full-width at half maximum (FWHM) Gaussian blur filter (*LANDSCAPE Consortium et al., 2016*; *LANDSCAPE Consortium et al., 2015*).

## Functional data analysis
Ten spherical a-priori-defined bilateral seed regions (*r* = 1 mm) were chosen based on the anatomical structures in order to compute functional connectivity maps of the motor system and the visual association system (reference) as follows: (1) primary motor system (seed voxels; MNI coordinates (x, y, z):±15, 30, 69), primary somatosensory cortex (±15,–30, 69), dorsolateral prefrontal association cortex (±22, 7, 55), Thalamus (±12,–29, 0), and visual association cortex (±51,–70, −13).

## Functional connectome reconstruction
We computed a functional connectome of the motor and visual association system using the defined seed regions by computing pairwise region-to-region functional connectivity and by computing Pearson's product moment correlation coefficient (*r*-value) pairwise between voxel time series. Correlation coefficient was Fisher's *r*- to *z*-transformed to normally distributed *z(r)* score for further statistical analysis.

## Interference statistics
Overall functional connectivity was computed and used as a regressor (*van den Heuvel et al., 2017*) prior to subjecting the data to a two-sided parametric Student's *t*-test in order to test for pairwise differences between ALS patients and controls using the network-based statistics approach (*Zalesky et al., 2010*).

## Acknowledgement
This work has been supported by the Baustein program of Ulm University Medical Faculty, by the Deutsche Forschungs Gemeinschaft (DFG) as part of the Graduate School in Cellular and Molecular Mechanisms in Aging and by the Synapsis Foundation. FR is also supported by the DFG as part of the Collaborative Research Center 1149 'Danger Response, Disturbance Factors and Regenerative Potential after Acute Trauma'. This study was also supported by the German Research Foundation (Deutsche Forschungsgemeinschaft, DFG Grant Number LU 336/15–1), and the German Network for Motor Neuron Diseases (BMBF 01GM1103A). The authors thank the Ulm University Center for

Translational Imaging MoMAN for its support. BC and DB are members of the International Graduate School in Molecular Medicine at Ulm University; DB is part of the Graduate School in Cellular and Molecular Mechanisms in Aging at Ulm University. The authors wish to acknowledge Mrs. Sonja Fuchs for MRI data acquisition of the human cohort; Ms. Silvia Cursano and Ms. Najwa Ouali Alami for the support and collaboration; Dr. Kim Moorwood and Mrs. Akila Chandrasekar for proofreading.

## Additional information

### Funding

| Funder | Grant reference number | Author |
| --- | --- | --- |
| Deutsche Forschungsgemeinschaft | GRK1789 | David Bayer |
| Deutsche Forschungsgemeinschaft | SFB1149 | Francesco Roselli<br>Tobias M Boeckers |
| Ulm University Medical Faculty | Baustein program | Francesco Roselli |

The funders had no role in study design, data collection and interpretation, or the decision to submit the work for publication.

### Author contributions

Barbara Commisso, Data curation, Formal analysis, Investigation, Visualization, Writing—original draft, Writing—review and editing; Lingjun Ding, Data curation, Investigation, Writing—original draft; Karl Varadi, Resources, Methodology, Writing—original draft; Martin Gorges, Resources, Data curation, Investigation, Visualization, Writing—original draft; David Bayer, Data curation, Investigation; Tobias M Boeckers, Resources, Supervision, Writing—original draft, Project administration, Writing—review and editing; Albert C Ludolph, Supervision, Funding acquisition, Writing—original draft, Project administration, Writing—review and editing; Jan Kassubek, Conceptualization, Resources, Supervision, Writing—original draft, Project administration, Writing—review and editing; Oliver J Müller, Resources, Supervision, Methodology, Writing—original draft, Writing—review and editing; Francesco Roselli, Conceptualization, Funding acquisition, Methodology, Writing—original draft, Project administration, Writing—review and editing

### Author ORCIDs

Francesco Roselli (iD) http://orcid.org/0000-0001-9935-6899

### Ethics

Human subjects: All subjects included in the human study provided written informed consent according to institutional guidelines; the consent includes the declaration of the understanding of the study design, the agreement to the participation to the study, to the publication of the results, and to the data protection and anonymization procedures (under the chaptes Einwilligungserklaerung", "Probandeninformation", "Darstellung der Experimente", "Datenschutzerklärung"). The study was approved by the Ethics Committee of Ulm University, Ulm, Germany (reference #19/12) and was performed in accordance with the ethical standards laid down in the 1964 Declaration of Helsinki and its later amendments.

Animal experimentation: All animal experiments have been approved by the Regierungpraesidium Tubingen under the licence no. 1312. All animals were handled according to the federal regulations on animal experimentations and under the supervision of the local veterinary office. Every effort was made to adhere to the 3R guidelines and to minimize suffering.

### Decision letter and Author response

Decision letter https://doi.org/10.7554/eLife.36892.036
Author response https://doi.org/10.7554/eLife.36892.037

# Additional files

## Supplementary files

• Supplementary file 1. (a) AAV variants injected in dorsal striatum (DS) and analyzed for their local infectivity ability. Number of neurons are normalized over AAV9- WT and expressed in terms of percentage. (b) AAV variants injected in dorsal striatum (DS) and analyzed for their retrograde infection ability to substantia nigra (SNr). Number of neurons are normalized over AAV9- WT and expressed in terms of percentage. (c) AAV variants injected in dorsal striatum (DS) and analyzed for their retrograde infection ability to motor cortex (MO). Number of neurons are normalized over AAV9- WT and expressed in terms of percentage. (d) AAV variants injected in primary visual cortex (V1) and analyzed for their retrograde infection ability to lateral geniculate nucleus (LGN). Number of neurons are normalized over AAV9- WT and expressed in terms of percentage. (e) Input to primary motor cortex (pMO) and secondary motor cortex (sMO) in WT animals (P20) traced via AAV9-SLR injection. Numbers are expressed in term of total neuronal count. (f) Input to primary motor cortex (pMO) and secondary motor cortex (sMO) in WT animals(P20) traced via AAV9-SLR injection. Neuronal count normalized for the volume of the injection site. (g) Input to primary motor cortex (pMO) and secondary motor cortex (sMO) in WT animals (P20) traced via AAV9-SLR injection. Total count was normalized for the volume of the injection site and contribution from each brain region is reported in term of percentage. (i) Input from thalamic nuclei to primary motor cortex (pMO) and secondary motor cortex (sMO) in WT animals(P20) traced via AAV9-SLR injection. Neuronal count normalized for the volume of the injection site. (j) Input from thalamic nuclei to primary motor cortex (pMO) and secondary motor cortex (sMO) in WT animals(P20) traced via AAV9-SLR injection. Total count was normalized for the volume of the injection site and contribution from each nucleus is reported in term of percentage.

DOI: https://doi.org/10.7554/eLife.36892.029

• Supplementary file 2. (a) Input to primary motor cortex in WT and mSOD animals (P20) traced via AAV9-SLR injection.Analysis of total number of neurons is reported. Numbers are expressed in term of total neuronal count. (b) Input to primary motor cortex in WT and mSOD animals (P20) traced via AAV9-SLR injection. Neuronal count normalized for the volume of the injection site. (c) Input from cortical layers (ipsilateral SS and ipsilateral AUD) to primary motor cortex in WT and mSOD animals (P20) traced via AAV9-SLR injection. Numbers are expressed in term of total neuronal count. (c) Input from cortical layers (ipsilateral SS and ipsilateral AUD) to primary motor cortex in WT and mSOD animals (P20) traced via AAV9-SLR injection. Neuronal count normalized for the volume of the injection site. (e) Input from cortical layers (ipsilateral SS and ipsilateral AUD) to primary motor cortex in WT and mSOD animals (P20) traced via AAV9-SLR injection. Total neuronal count was normalized for the volume of the injection site, values are expressed in terms of percentage. (f) Input to primary motor cortex in WT and mSOD animals (P20) traced via choleratoxin (CTb) injection. Numbers are expressed in term of total neuronal count. (g) Input to primary motor cortex in WT and mSOD animals (P20) traced via choleratoxin (CTb) injection. Neuronal count normalized for the volume of the injection site. (h) Input to primary visual cortex (V1) in WT and mSOD animals (P20) traced via AAV9-SLR injection. Numbers are expressed in term of total neuronal count. (i) Input to primary visual cortex (V1) in WT and mSOD animals (P20) traced via AAV9-SLR injection. Neuronal count normalized for the volume of the injection site. (j) Input to secondary motor cortex in WT and mSOD animals (P20) traced via AAV9-SLR injection. Analysis of total number of neurons is reported. Numbers are expressed in term of total neuronal count. (k) Input to secondary motor cortex in WT and mSOD animals (P20) traced via AAV9-SLR injection. Neuronal count normalized for the volume of the injection site.

DOI: https://doi.org/10.7554/eLife.36892.030

• Supplementary file 3. (a) Input to primary motor cortex in mSOD animals traced via AAV9-SLR injection during disease progression: early pre- symptomatic(P20), intermediate (P60) and later stage (P90). Total neuronal count was normalized for the volume of injections ite, numbers expressed in term of fold change over the relative WT. (b) Spine density analysis on basal dendrites (10 μm stretch) of cortical pyramidal neurons projecting to pMO. Comparison between WT and earlypre-symptomatic mice (P20). Tracing via AAV9-SLR injected in pMO. (c) Spine density analysis on basal dendrites (10 μm stretch) of cortical pyramidal neurons projecting to pMO. Comparison between

WT and intermediate stage mice (P60). Tracing via AAV9-SLR injected in pMO. (d) Spine density analysis on basal dendrites (10 μm stretch) of cortical pyramidal neurons projecting to pMO. Comparison between WT and later stage mice (P90). Tracing via AAV9-SLR injected in pMO. (e) Analysis for misfSOD intensity in cortical projecting neurons to primary motor cortex in mSOD pre-symptomatic mice (P20). Tracing via choleratoxin (CTb). (f) Analysis for misfSOD intensity in projecting neurons to SS in mSOD mice (P50). Tracing via choleratoxin (CTb). (g) Spine density analysis on basal dendrites (10 μm stretch) of cortical pyramidal neurons projecting to primary motor cortex. Comparison between misfSOD- and misfSOD +neurons of mSOD mice (P60). Tracing via AAV9-SLR injected in pMO.

DOI: https://doi.org/10.7554/eLife.36892.031

• Transparent reporting form

DOI: https://doi.org/10.7554/eLife.36892.032

## Data availability

All the murine data generated or analysed during this study are included in the manuscript and supporting files. The raw images are deposited on the Dataverse database (https://doi.org/10.7910/DVN/5VNSXE)

The following dataset was generated:

| Author(s) | Year | Dataset title | Dataset URL | Database, license, and accessibility information |
|---|---|---|---|---|
| Barbara Commisso | 2018 | Replication Data for: Stage-dependent remodeling of large-scale architecture of projections to primary motor cortex in ALS mouse model revealed by a new variant retrograde-AAV9 | https://doi.org/10.7910/DVN/5VNSXE | Publicly available at the Harvard dataverse (DOI: 10.7910/DVN/5VNSXE) |

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
