## [Decision Letter]

Thank you for submitting your article "Stage-dependent hyperconnectivity to primary motor cortex in ALS mouse model revealed via retrograde- AAV9" for consideration by *eLife*. Your article has been reviewed by four peer reviewers, including Moritz Helmstaedter as the Reviewing Editor and Reviewer #1, and the evaluation has been overseen by Richard Ivry as the Senior Editor. The following individuals involved in review of your submission have agreed to reveal their identity: Gretchen Thomsen (Reviewer #3); Edward Stern (Reviewer #4).

The reviewers have discussed the reviews with one another and the Reviewing Editor has drafted this decision to help you prepare a revised submission.

This manuscript reports changes in the amounts of labeled axons projecting to the primary motor area in a mouse model of ALS and combines these results with fMRI data from human patients. This work emphasizes the need to consider the upper motor neurons in the SOD model of ALS.

While the reviewers were positive about the work, they agreed on a list of important issues that would need to be addressed in the revised manuscript:

1) Validity of the correspondence claims between Human and mouse data (see comments by reviewer #2 below).

2) Methodological caveats with interpreting the viral tracing data: need for reporting of absolute numbers, blinding of the analysis (see comments by all reviewers); comparison to other methods, justification of development of a new virus compared to e.g. Tervo et al., 2016.

3) General overstatement of "connectivity" while the experiments only provided axonal labeling. Here we would request a condensed revision that avoids overstatements as to the equivalence of "connectivity" with the number or density of axonal projections.

4) Statistical concerns (especially reviewer #4).

5) Lack of discussion of the limitations of the chosen tracing method: inability to analyse interneuron projections, lack of functional correlates (here we do not consider additional experiments necessary, but a more careful presentation and extended discussion.)

6) Relevance/interpretation of the spine data (Figure 6, see reviewer #2).

Reviewer #1

1) First, a key caveat of viral axonal tracing is the rate of infection. In order to allow a serious judgment of the data it is therefore critical to report raw neuron numbers first before a further processing of the data in terms of fractions, normalized numbers etc. can take place. Here, Figure 2 does not currently allow judging of the number of labeled presynaptic neurons. None of the images provides a resolution that allows counting neurons. The figure legend does not allow understanding of what is shown in panel B. Is this actual data, (i.e. numbers of neurons?) or just a sketch. The fractions reported in all panels are probably percent, this label is missing but here it is excessively important to report total numbers.

2) Furthermore, when the authors turn to the effect in the disease model, Figure 3, they switch to a different reporting axis, namely normalized neuron count. This is not clear why this is necessary, I understand that this is a normalization to the injection volume, but it leaves a certain amount of doubt whether this was necessary compared to reporting fractions of neurons as done before. I think it is necessary to first report all the raw numbers and then make very clear how the further processing of these numbers took place. Especially, since the effect is very strong and surprising as shown in Figure 3A, it is very important to understand whether this is driven by primarily the raw data or rather further enhanced by the normalization procedure that was used.

3) Third, the problem that figure legends are not overly informative and figure axes are sometimes not properly labeled is prevalent throughout the manuscript, for instance Figure 2—figure supplement 1, it is not made clear that this is likely the percentage of neurons from thalamus, otherwise the numbers wouldn't add up again (percent sign is missing). Again, absolute numbers would be very helpful together with more raw data images at high resolution as to allow the reader to judge the raw data in this work.

4) In the analysis of axonal projections from PMO, Figure 8, the analysis is carried out by reporting axonal volume. While this a plausible measure because it can be estimated from images, it is not extremely informative in quantitative terms. Furthermore, there are substantial issues in how to define the volume outlines giving different intensity levels of images. Here, at least a more thorough justification of this quantification and an attempt to relate this to the amount of connectivity would be needed. It is very unlikely that axonal volume directly correlates with the number of target neurons. Since the manuscript is otherwise quantitative about projections at the single neuron level it is surprising to the reader that this measure is of such coarse resolution here.

5) In this setting of small n of animals (3 versus 6) it is very important that the counting of neurons was blinded to the disease condition. I was not able to see in the methods that this was in fact carried out in a blinded fashion (the statement reads "the experimenter was kept blind to the genotype till the analysis stage", Materials and methods section – does this include the image analysis, i.e. neuron counting, injection volume normalization etc.?). It should include these steps.

Reviewer #2

1) I am not sure why the authors claim, "the motor subnetwork in an ALS mouse model displays selective increase in projections from SS and sMO to pMO in a pattern that closely matches the connectivity measurement by fMRI in human patients". While they find alterations in anatomical projection densities in the mouse between specific brain regions (most specifically somatosensory and contralateral motor cortices), in humans the find increases in functional connectivity strength between the primary somatosensory cortices and between pMO and some prefrontal cortex areas. Perhaps there are analogies between brain regions that I do not know about, but the authors should discuss in how far these patterns "closely match". Perhaps, side-by-side schematics of pMO inputs and their relative strength under normal and ALS conditions in mice and humans can clarify what is "closely match[ing]" here.

2) The presence of an increased retrograde labeling density does not necessarily imply any specific functional pattern of connectivity – for example, it is well documented, e.g. in the peripheral motor system, that axonal projections that are chronically silenced would show sprouting, which could easily result in increased retrograde tracing without any accompanying increase in "functional connectivity". Thus, the functional predictions that can be supported by an anatomical pattern of connectivity are typically weak. In general, while I am a bit skeptical that it will be easy to "interpret" fMRI findings in sporadic human ALS based on data from the mouse SOD model; at least similar modalities should be compared, i.e. structural data obtained from mice and humans or functional measurements dine in both species – ideally both kinds of measurements in parallel. This is clearly not a (reasonable) revision request but illustrates my general doubt whether it is meaningful to combine widely disparate types of data from a specific genetic animal model of disease and from sporadic human patients.

3) I think the term "connectomics" should be avoided here – at least for the mouse no evidence of any "connectivity" is provided, and certainly, no "comparison at circuit level of ALS mouse model and human connectomics" was attempted or achieved.

4) I was not sure what purpose the spine analysis in input fields of pMO in Figure 6 served (it was bit opaque to me what the difference is between what is shown in Figure 5 vs. Figure 6A – the authors need to clarify what the difference is or combine in a meaningful way). These data are interesting in their own right, especially if in a cell-to-cell comparison, dissociation between spine loss and misfolded SOD content could be shown. However, I do not see what they explain here, especially as the patterns between local spine density and projection strength do not seem to correlate especially well. I wonder whether this belongs into another "story".

Reviewer #3

1) The interpretation of results is a little bit unclear because of the complication that the quantification of labeled cells by means of anterograde or retrograde infectivity depends on the number of cells originally taking up the virus. If there are smaller numbers of cell populations in the injected region due to SOD1 disease pathology, there will be fewer projection neurons labeled. Is there altered connectivity, or just fewer cells in the injection regions delivering the virus?

2) I am curious as to why there were no endpoint studies conducted and the latest timepoint was 90/105 days. This seems an obvious piece of information that is missing to observe connectivity encompassing the entire spectrum of disease.

Reviewer #4

1) I found the study to be extremely well done. However, I have some issues with the statistical analysis. In general, the authors are quite rigorous. They correctly set the p (α) value prior to the analysis and determine significance accordingly. However, they then proceed to ascribe larger significance levels to the results of various comparisons, which is simply incorrect. The p value is the one which was set a priori: in this case 0.05. However, the authors are correct in trying to emphasize stronger effects. For this, I strongly recommend leaving the p values at 0.05, starred with an asterisk as done in the paper, and adding a statistic called Effect Size. There are several ways of calculating this, and the authors need pick the most appropriate (with, of course, citation of the exact method used in the methods section). These calculations will not take much time, and, in my opinion, greatly improve the paper. Especially for papers with great clinical significance, as I believe this study to be, this statistic is of increasing importance.

A good article detailing the procedures can be found here:https://www.ncbi.nlm.nih.gov/pmc/articles/PMC3444174/

2) Another issue that concerns me is the specific types of changes in the distributions of the projections to and from the motor areas. The authors use mean and standard deviations throughout the paper, which, though reasonable, leave open a few questions. The entire study is about changes in the distributions of the projections. Mean and standard deviation alone are adequate, depending on the type of distributions. To give an idea, the authors should include the medians alongside the means. Usually, of course, the statistics will be similar. But, for example, if the medians are consistently above the means, that also tells us something about the shape of the distributions of the projections. And if that effect grows or shrinks in the Tg cases, that tells us about the effect of the pathology on the distributions. Of course, there are other methods for more sophisticated analysis of the changes in the distributions. Calculations of skewness and kurtosis come to mind. But this would be a method that yields a great deal of information without too much additional time and effort on the part of the authors. I emphasize that even if the distributions are completely normal, with no significant differences between means and medians, these statistics should be included in the paper: the information is of value.

3) Figure 3D: Regarding the reduction in percentage of projecting neurons of the SOD mice: While the changes in the mean are clear, a feature that seems to be overlooked is the reduction in variability in the Tg samples. While the statistical significance of the effect is not in doubt, the reason for the effect is of interest. The reduction in SD looks greater than one would expect from the properties of the distribution. I would ask the authors to include a table of coefficients of variation (SD/mean) of the 24 groups in the figure. Perhaps a statistical comparison of the CVs would also be appropriate. If there is a significant lowering of the CV in some the SOD cortex groups, this means that a significant reduction of the variability of this group is present. This might mean that some particular subpopulation of projections is eliminated, as opposed to a random reduction.

---

## [Author Response]

Reviewer #11) First, a key caveat of viral axonal tracing is the rate of infection. In order to allow a serious judgment of the data it is therefore critical to report raw neuron numbers first before a further processing of the data in terms of fractions, normalized numbers etc. can take place. Here, Figure 2 does not currently allow judging of the number of labeled presynaptic neurons. None of the images provides a resolution that allows counting neurons. The figure legend does not allow understanding of what is shown in panel B. Is this actual data, (i.e. numbers of neurons?) or just a sketch. The fractions reported in all panels are probably percent, this label is missing but here it is excessively important to report total numbers.

We have followed the recommendations of the reviewer and thoroughly revised the manuscript and the data presented. In detail:

- We now present the raw number of projecting neurons to pMO or sMO in Figure 2; we had previously reported only the fraction (% of total number of neurons projecting to MO) represented by each structure in order to highlight their relative contribution to the total input to MO structures. We have now moved the graph depicting the fractions to the corresponding supplementary figure, in keeping with the suggestion to depict the raw numbers first.

- We now provide new representative pictures for Figure 2: a low-magnification overview picture of the brain section, accompanied by several high-resolution confocal panels depicting specific regions of interest (thalamic nuclei, primary somatosensory, contralateral motor cortex); in the high-resolution images single neurons can be easily identified.

- The panel B was intended to be a graphical depiction of the experimental design (injection site, regions in which projecting neurons have been analyzed) rather than a quantitative summary of the experiment. We have maintained the graphical depictions, clearly stating their purpose, and we have added small tables summarizing the experimental details. In a set of supplementary tables now we present the raw data corresponding to the normalized data represented in the main figure, to allow the comparison; all trends detectable in the raw-counts graphs are also detectable in the normalized-counts graphs although in the former, being the dataset more noisy (since one source of variability, namely the size of the injection site, has not been removed), some differences do not reach statistical significance. In addition, we provide the numerical data used to derive each graph as accompanying excel files, and we also provide the statistical details (mean, median, standard deviation, skewness, effect size, coefficient of variation and statistical significance) for each panel as supplementary tables.

- We have expanded the labels of every graph to include more details about the type of data depicted (raw counts, percentages, normalized counts) and we have stressed the type of processing (raw counts, normalized counts) also in the figure legends.

2) Furthermore, when the authors turn to the effect in the disease model, Figure 3, they switch to a different reporting axis, namely normalized neuron count. This is not clear why this is necessary, I understand that this is a normalization to the injection volume, but it leaves a certain amount of doubt whether this was necessary compared to reporting fractions of neurons as done before. I think it is necessary to first report all the raw numbers and then make very clear how the further processing of these numbers took place. Especially, since the effect is very strong and surprising as shown in Figure 3A, it is very important to understand whether this is driven by primarily the raw data or rather further enhanced by the normalization procedure that was used.

In the first version of the manuscript, we displayed the fraction of the total population of neurons projecting to pMO and sMO in Figure 2 to better depict the relative contribution of each structure to the total input, and we switched to normalized counts in Figure 3 to comparatively show that the number of neurons projecting to pMO was different in mSOD vs WT. We have now revised all figures to depict normalized values in the main figures and we provide the raw-counting (before normalization) in corresponding supplementary figures; for Figure 2, we also provide supplementary graphs depicting, for each structure, the fractional contribution to the overall pool of neurons projecting to pMO or sMO. In addition, we provide not supplementary tables in which the actual values (both before and after normalization), together with the statistical descriptors and the statistical significance, are reported. Because variability in the volume of the injection site can be minimized under the best experimental procedures but cannot be eliminated (as also shown in the dataset of the Allen Brain Institute), normalization for the injection volume, as performed by Oh et al., 2014, has been used to reduce this source of variability in the dataset. The increased number of neurons projecting from SS to pMO is already clearly detectable in the raw data and remains statistically significant after injection volume normalization and, with the exception of the increase in projections from cMOs to pMO at P30 (which is significant only upon normalization), all other differences are seen in the raw data as much as in normalized ones.

3) Third, the problem that figure legends are not overly informative and figure axes are sometimes not properly labeled is prevalent throughout the manuscript, for instance Figure 2—figure supplement 1, it is not made clear that this is likely the percentage of neurons from thalamus, otherwise the numbers wouldn't add up again (percent sign is missing). Again, absolute numbers would be very helpful together with more raw data images at high resolution as to allow the reader to judge the raw data in this work.

We have now thoroughly revised the figure legends and the details of the figures to ensure that the best description of the experimental design, of the results and of the graphs is achieved. In particular, for each graph legend we make clear the type of data depicted and the rationale for the use of that parameter; for the majority of the graphs, we provide now (in supplementary figures and tables) raw values, normalized values and/or percentages.

In the case of Figure 2—figure supplement 1, having presented in Figure 2 the share of neurons projecting to pMO located in thalamus, we broke down the number to highlight the contribution of each thalamic nucleus to that share; we have now provided both the raw number of neurons and the percentage of the total thalamic projection to pMO.

We have now added or substituted images using high-resolution confocal representative images.

4) In the analysis of axonal projections from PMO, Figure 8, the analysis is carried out by reporting axonal volume. While this a plausible measure because it can be estimated from images, it is not extremely informative in quantitative terms. Furthermore, there are substantial issues in how to define the volume outlines giving different intensity levels of images. Here, at least a more thorough justification of this quantification and an attempt to relate this to the amount of connectivity would be needed. It is very unlikely that axonal volume directly correlates with the number of target neurons. Since the manuscript is otherwise quantitative about projections at the single neuron level it is surprising to the reader that this measure is of such coarse resolution here.

In the initial version of the manuscript, we intended this piece of information to integrate the options for the interpretation of the fMRI-derived hyperconnectivity. However, we agree with the reviewer that, because of the intrinsic difference in the way anterograde and retrograde tracing experiments are quantified, the degree of resolution in the values cannot be compared. We therefore agree with the reviewer´s opinion that the dataset is more homogeneous, quantitative and streamlined without the anterograde tracing experiments and we have removed them from the revised version of the manuscript now resubmitted. We reserve to expand further the dataset on anterograde tracing in further studies. We have revised the text and the figures accordingly.

5) In this setting of small n of animals (3 versus 6) it is very important that the counting of neurons was blinded to the disease condition. I was not able to see in the methods that this was in fact carried out in a blinded fashion (the statement reads "the experimenter was kept blind to the genotype till the analysis stage", Materials and methods section – does this include the image analysis, i.e. neuron counting, injection volume normalization etc.?). It should include these steps.

The genotyping of the mouse was performed by a researcher different from the one performing the viral injection, the histology, the imaging and the image quantification, which includes the counting of all neurons. The genotype of each mouse was revealed only once the counting of the neurons and their anatomical annotation was closed and the corresponding file transferred to a different computer for the statistical analysis. At that point, individual mice were grouped according to the genotype. Incidentally, this design is the reason why the two groups have different n (6 vs 3), since mice were not pre-selected based on their genotype. We have now added more details to the description of the blinding design in the Materials and methods section.

Reviewer #21) I am not sure why the authors claim, "the motor subnetwork in an ALS mouse model displays selective increase in projections from SS and sMO to pMO in a pattern that closely matches the connectivity measurement by fMRI in human patients". While they find alterations in anatomical projection densities in the mouse between specific brain regions (most specifically somatosensory and contralateral motor cortices), in humans the find increases in functional connectivity strength between the primary somatosensory cortices and between pMO and some prefrontal cortex areas. Perhaps there are analogies between brain regions that I do not know about, but the authors should discuss in how far these patterns "closely match". Perhaps, side-by-side schematics of pMO inputs and their relative strength under normal and ALS conditions in mice and humans can clarify what is "closely match[ing]" here.

We have now detailed further our statement regarding (avoiding the use of the potentially misleading wording “closely matches” and referring to the human and mouse data as “sharing similarities”) the match between mouse and human data, underscoring that: (i) both in human and mouse data, we observe a progressive remodeling of the motor subnetwork (ii) both in (previously published) human data and current mouse data, the remodeling of the motor subnetwork is detectable much before the appearance of overt clinical or behavioural abnormalities (iii) both in human and mouse data, the remodeling of the motor subnetwork involves the primary motor cortex and more rostral, homolateral or controlateral structures (in the frontal lobe, prefrontal cortex or secondary motor cortex in human and mouse, respectively) and projecting to pMO iv) both in human and mouse data the primary somatosensory cortex appears involved in the motor subnetwork remodeling (either in terms of increased projection to pMO in the mouse, or in terms of inter-SS connectivity, connectivity in the sensorymotor cortex (close proximity prevents precise delineation of SS from pMO) and SS connectivity to other cortical areas (Agosta et al., 2011; Douaud et al., 2011); admittedly, the inter-SS projections were not investigated in our mouse dataset. Nevertheless, divergences were also observed, namely in the lack of hyperconnectivity from thalamus to pMO in ALS patients.

We agree with the reviewer that precise analogies between different cortical areas in mouse and humans are not always established, and we have now amended the discussion to include a detailed comparison of human and mouse data, highlighting the similarities and the possible divergences.

2) The presence of an increased retrograde labeling density does not necessarily imply any specific functional pattern of connectivity – for example, it is well documented, e.g. in the peripheral motor system, that axonal projections that are chronically silenced would show sprouting, which could easily result in increased retrograde tracing without any accompanying increase in "functional connectivity". Thus, the functional predictions that can be supported by an anatomical pattern of connectivity are typically weak. In general, while I am a bit skeptical that it will be easy to "interpret" fMRI findings in sporadic human ALS based on data from the mouse SOD model; at least similar modalities should be compared, i.e. structural data obtained from mice and humans or functional measurements dine in both species – ideally both kinds of measurements in parallel. This is clearly not a (reasonable) revision request but illustrates my general doubt whether it is meaningful to combine widely disparate types of data from a specific genetic animal model of disease and from sporadic human patients.

We accept the suggestion of the reviewer and we now provide a more balanced presentation of the goals that can be obtained by our work, not in terms of “interpreting” fMRI data but rather in terms of demonstrating that the large-scale circuit remodeling identified in functional human data can be detected in high-resolution structural data obtained in the mouse model and that the occurrence of the remodeling and similarities in the pattern can be drawn. Our data show that the motor subnetwork in the ALS model undergoes a progressive remodeling during disease progression, as also shown in fMRI data and therefore provide ground to further explore this phenomenon using other modalities.

Regarding the connection between structural and functional remodeling, we agree that this correlation is not always perfect. However, several studies have shown that the structure of cortical networks provides a strong template for functional connectivity: structurally connected regions are known to display high levels of functional connectivity, e.g. in the default brain network (Honey et al., 2007; Greicius et al., 2009) and the functional and structural connectivity maps are highly correlated (van den Heuvel et al., 2009). At large scale, a strong correlation between the degree of structural connectivity (defined by DTI measurements in white matter tracts) and of functional connectivity has been reported for the resting state network (Hagmann et al., 2008) as well as for resting and task-specific networks in aggregated as well as in single-individual data (Hermundstadt et al., 2013). Furthermore, the combination of anatomical tracing with fMRI measurements in non-human primates has shown that functional connectivity increases with structural connectivity (Shen et al., 2012) and the temporal stability of functional connectivity is highly correlated with the structural connectivity (Shen et al., 2015). On the other hand, functional or physiological data do not necessarily imply structural connectivity, since indirect connections may contribute to it, at least in part (Adachi et al., 2012). Thus, we now discuss the mouse findings, within the limits of extrapolating murine data to humans, as indicating one of the possible mechanisms involved in the genesis of hyperconnectivity, since structural connectivity may constraint functional connectivity pattern.

We also discuss the expansion of projection to pMO as non-mutually exclusive with other mechanisms that may involve in the increased connectivity (as seen in fMRI data). In particular, we discuss also the functional or anatomical dysfunction of inhibitory interneurons (Kew et al., 1993; Douaud et al. 2013) as a potential mechanism.

We have followed the reviewer´s suggestion on the mechanistic interpretation of the increased projection origin. In fact, we believe, in agreement with the he reviewer´s mention of the neuromuscular unit as model, that the peripheral motoneuron-muscle innervation may be a useful example in modeling sprouting: in fact, silencing of muscle cells (as obtained by blocking the neuromuscular junction with botulinum toxin or as occurring during denervation-reinnervation phenomena) leads to sprouting of the motoneurons projecting to the these cells through mechanisms driven to large extent (although not only) by the denervated cells (Caroni et al., 1994; Caroni et al., 1997); under the condition of denervation-induced sprouting, however the functional response of the muscle (since the definition of functional connectivity, which is based on the correlation of activity pattern, cannot be easily applied to the peripheral nervous system) is indeed altered, as shown by traditional emg studies, with a larger number of muscle cells recruited by the same motor neuron (and by changes in the temporal dynamics of compound motor unit potentials); so in the peripheral nervous system, increased sprouting resulting from reduced pre or post-synaptic activity leads to detectable changes in functional properties of the motor unit. We agree with the reviewer that our experimental data have not investigated whether the increase in projections originates as a consequence of post- or pre-synaptic abnormalities or may (unlikely) involve extensive synaptic silencing, a subject deserving a dedicated investigation; we now mention these mechanistic hypotheses in the Discussion section.

Moreover, we have now provided in the Introduction a more balanced description of the goals of our investigation in terms of the comparison of mouse vs human data, underscoring that, although the correlation between structural and functional connectivity is strong and repeatedly reported, the structural component is one of several factors determining the functional connectivity, and specifying that extrapolation from experimentally tractable models (such as the murine one) to humans must be accepted with caution.

3) I think the term "connectomics" should be avoided here – at least for the mouse no evidence of any "connectivity" is provided, and certainly, no "comparison at circuit level of ALS mouse model and human connectomics" was attempted or achieved.

We have amended the text to avoid reference to connectomics and we have re-phrased our statement to clarify that our data contribute to analysis of large-scale architecture of motor circuits in ALS mouse model and provide entry points to understand changes occurring in human patients.

4) I was not sure what purpose the spine analysis in input fields of pMO in Figure 6 served (it was bit opaque to me what the difference is between what is shown in Figure 5 vs. Figure 6A – the authors need to clarify what the difference is or combine in a meaningful way). These data are interesting in their own right, especially if in a cell-to-cell comparison, dissociation between spine loss and misfolded SOD content could be shown. However, I do not see what they explain here, especially as the patterns between local spine density and projection strength do not seem to correlate especially well. I wonder whether this belongs into another "story".

We now clarify that, having demonstrated the increased projections from SS and other cortical (and subcortical) structures at different stages of disease progression, we investigated if this phenotype was a consequence of pathogenic processes unfolding in the neurons projecting to pMO (i.e., if the neurons projecting from SS to pMO would display any sign of ongoing pathology which could be the driver of the increase in projections). We elected to monitor two readouts, namely the burden of misfolded SOD1 and the loss of spines, to determine if any pathology in the neurons projecting to pMO could justify the observed phenotype. The two readouts were selected in order to minimize the assumptions on the pathogenic process involved: whereas the burden of misfolded SOD1 appears to be very close to initial steps of the pathogenic cascade (the mutation of the very same gene is indeed pathogenic), the reduced spine density has been repeatedly identified in several neuronal subpopulations in ALS (e.g., Ozdinler et al., 2011; Fogarty et al., 2016, 2017) and offers a marker of the structural integrity of the neuron (which may be the result of the convergence of a number of ALS-related pathogenic processes, such as trafficking abnormalities, disturbed autophagy, altered cytoskeleton). We have demonstrated that the loss of spines does take place in the neurons projecting to pMO but there is no correlation between the increase in projections to pMO and the appearance of this sign of disease, suggesting that the increased-projection phenotype could not be a direct consequence of abnormalities (within the limits explored by the spine-loss readout) in projecting neurons. Of note, whereas loss of spines (in layer V neurons) was reported to occur in SS only at very advanced stages of the disease (Fogarty et al., 2016), in our subpopulation of neurons projecting to pMO this has been detected at earlier stages, underscoring the possibility of differential vulnerability of neuronal populations identified by their projections.

Regarding the specific points raised:

- We now present the data of Figure 6A first (neurons projecting to primary MO) and we have removed the data formerly in Figure 5 (neurons projecting to secondary MO) in order to focus on pMO data and streamline the presentation of the data.

- We now provide new data regarding the correlation between misfSOD burden and dendritic spine density (Figure 7 and Figure 8). We identified the subpopulation of pSS neurons projecting to pMO by injecting the latter with the retrograde AAV9 virus, and we assessed the burden of misfolded SOD1 by immunostaining; among the neurons projecting to pMO (ZsGreen+), we identified neurons with high misfSOD1 burden (fluorescence intensity>2000) and neurons with low misfSOD1 burden (fluorescence intensity < 1000). When we assessed the dendritic spines density for these two groups, we found that they were comparable and, in particular, the dendritic spine density in neurons in layer V (both in high-burden and in low-burden neurons) was significantly smaller than in layer II/III neurons (either high-burden either low-burden). Thus, these findings suggest that the loss-of-spines phenotype is dissociated from the misfolded SOD1 burden.

Reviewer #31) The interpretation of results is a little bit unclear because of the complication that the quantification of labeled cells by means of anterograde or retrograde infectivity depends on the number of cells originally taking up the virus. If there are smaller numbers of cell populations in the injected region due to SOD1 disease pathology, there will be fewer projection neurons labeled. Is there altered connectivity, or just fewer cells in the injection regions delivering the virus?

The determination of the number of distant neurons projecting to a given volume injected with the AAV suspension is allowed by the viral infection of the axons contained within the injection volume and it is therefore not directly dependent on the number of neurons whose cell body is located within the injection volume. In contrast, the number of target neurons located within the injection volume is a critical factor in experiments using monosynaptic-retrograde viruses such as replication-incompetent rabies (rabied-DeltaG). In our system, the density of neurons in the motor cortex would not affect the infectivity or the retrograde transport of the AAV per se, unless a severe decrease in the cell number in the injected region would cause the massive loss of projecting axons, due to the degeneration of their postsynaptic counterparts. However, we have consistently observed an expansion in the number of neurons (therefore, axons) projecting to the injection volume, indicating that a net loss of axons was not taking place. Loss of neurons may, speculatively, generate conditions that may increase axonal sprouting. We have now verified, using NeuN staining, that the neuronal density in pMO is comparable in WT and mSOD1 mice at least up to P90. Therefore, regarding projection to pMO, we believe that the changes in the number of projecting neurons are not due, or are not related, to loss of neurons in pMO. We have added the data relative to overall neuronal density in pMO in Figure 6—figure supplement 1 and we have amended the text accordingly.

2) I am curious as to why there were no endpoint studies conducted and the latest timepoint was 90/105 days. This seems an obvious piece of information that is missing to observe connectivity encompassing the entire spectrum of disease.

We agree with the reviewer that an evaluation at end-stage would have provided additional information; in order to maintain the same interval between injection and sacrifice as in other timepoints, mice for this end-stage assessment are to be injected at P115-120 and sacrificed at P130-135 (approximatively the anticipated average survival for this strain of SOD1(G93A) mice); at P115-120 this mice already display a severe motor impairment. In our preliminary investigation, we subject to intracerebral injections 3 SOD1(G93A) mice at P117, and none of them survived the procedure because of respiratory failure during the isoflurane anesthesia. Therefore, we believed unethical to proceed with experiments at timepoints in which very high mortality rate could be expected. We have added an explanation for the lack of a more advanced timepoint to the Results section.

Reviewer #41) I found the study to be extremely well done. However, I have some issues with the statistical analysis. In general, the authors are quite rigorous. They correctly set the p (α) value prior to the analysis and determine significance accordingly. However, they then proceed to ascribe larger significance levels to the results of various comparisons, which is simply incorrect. The p value is the one which was set a priori: in this case 0.05. However, the authors are correct in trying to emphasize stronger effects. For this, I strongly recommend leaving the p values at 0.05, starred with an asterisk as done in the paper, and adding a statistic called Effect Size. There are several ways of calculating this, and the authors need pick the most appropriate (with, of course, citation of the exact method used in the methods section). These calculations will not take much time, and, in my opinion, greatly improve the paper. Especially for papers with great clinical significance, as I believe this study to be, this statistic is of increasing importance.A good article detailing the procedures can be found here:https://www.ncbi.nlm.nih.gov/pmc/articles/PMC3444174/

We thank the reviewer for his suggestions. We have amended the graphs as recommended, highlighting with one asterisk the differences that are significant, irrespective of the value of p as far as p<0.05. In order to underscore the magnitude of the effects observed, we now report the effect size (calculated according to Cohen´s d approach) in a tabulated format so that for each structure whose projections to pMO are compared in WT and mSOD, both the Effect Size value and the statistical significance is available. In keeping with the requests of other reviewers, we provide the Effect Size value only in the tables in order to streamline the text and improve readability.

2) Another issue that concerns me is the specific types of changes in the distributions of the projections to and from the motor areas. The authors use mean and standard deviations throughout the paper, which, though reasonable, leave open a few questions. The entire study is about changes in the distributions of the projections. Mean and standard deviation alone are adequate, depending on the type of distributions. To give an idea, the authors should include the medians alongside the means. Usually, of course, the statistics will be similar. But, for example, if the medians are consistently above the means, that also tells us something about the shape of the distributions of the projections. And if that effect grows or shrinks in the Tg cases, that tells us about the effect of the pathology on the distributions. Of course, there are other methods for more sophisticated analysis of the changes in the distributions. Calculations of skewness and kurtosis come to mind. But this would be a method that yields a great deal of information without too much additional time and effort on the part of the authors. I emphasize that even if the distributions are completely normal, with no significant differences between means and medians, these statistics should be included in the paper: the information is of value.

We have now enriched the statistical analysis by providing the value of the median along with the mean for every structure considered, both for raw and normalized data; in order to improve readability and enable an easier comparison, we have provided the values in tabular format along with mean and standard deviation. The median values appear to be, with the exclusion of areas with small number of projecting neurons, very close (in most cases within 10%) of the mean value. In order to properly characterize the distribution, we have also computed the skewness for each group and we have tabulated the values along the mean, the median, the standard deviation and other statistical parameters. The skewness value was often close to 0 (in particular for structures with large number of projecting neurons) and in no case exceeding the value of ± 2.0 (the largest value observed being 1.7 for perirhinal cortex, in which only 25 pMO-projecting neurons were identified), indicating a limited impact of non-normal distribution on the results reported. We have amended the text in the methods section to provide a description of the calculation performed and a short summary of the interpretation of the Skewness values, and we have mentioned the median and skewness data in the Results section.

3) Figure 3D: Regarding the reduction in percentage of projecting neurons of the SOD mice: While the changes in the mean are clear, a feature that seems to be overlooked is the reduction in variability in the Tg samples. While the statistical significance of the effect is not in doubt, the reason for the effect is of interest. The reduction in SD looks greater than one would expect from the properties of the distribution. I would ask the authors to include a table of coefficients of variation (SD/mean) of the 24 groups in the figure. Perhaps a statistical comparison of the CVs would also be appropriate. If there is a significant lowering of the CV in some the SOD cortex groups, this means that a significant reduction of the variability of this group is present. This might mean that some particular subpopulation of projections is eliminated, as opposed to a random reduction.

We have now presented the data in Figure 3D in histogram format, in order to make the value of each column more easily compared, and we are providing all the numerical values, together with the value of the coefficient of variation, in a companion table. However, with the exclusion of the neurons allocated to layer iv (between 0 and 1.5%), the CV for all other cortical layers was comparable, preventing further analysis in this direction.